# It Ain't Over: A Multi-aspect Diverse Math Word Problem Dataset

**Jiwoo Kim    Youngbin Kim    Ilwoong Baek    JinYeong Bak    Jongwuk Lee** [*]

Sungkyunkwan University, Republic of Korea

{jindog1210, andyk3603, alltun100, jy.bak, jongwuklee}@skku.edu

## Abstract

The math word problem (MWP) is a complex task that requires natural language understanding and logical reasoning to extract key knowledge from natural language narratives. Previous studies have provided various MWP datasets but lack diversity in problem types, lexical usage patterns, languages, and annotations for intermediate solutions. To address these limitations, we introduce a new MWP dataset, named *DMath* (**D**iverse **Math** Word Problems), offering a wide range of diversity in problem types, lexical usage patterns, languages, and intermediate solutions. The problems are available in English and Korean and include an expression tree and Python code as intermediate solutions. Through extensive experiments, we demonstrate that the DMath dataset provides a new opportunity to evaluate the capability of large language models, *i.e.*, GPT-4 only achieves about 75% accuracy on the DMath[1] dataset.

## 1 Introduction

The math word problem (MWP) is a challenging and intriguing task that requires a deep understanding of natural language narratives and logical reasoning (Bobrow, 1964; Kushman et al., 2014). The narratives typically involve several numbers and unknown variables. Also, some problems demand commonsense reasoning. Solving these problems requires a comprehensive understanding of world knowledge and the reasoning between mathematical representations within a given context.

As the first step to developing MWP models, it is vital to collect a representative subset of MWP to construct MWP benchmark datasets. Although many existing studies (Koncel-Kedziorski et al., 2016; Roy and Roth, 2017; Miao et al., 2020) have published various MWP corpora, they mostly lack

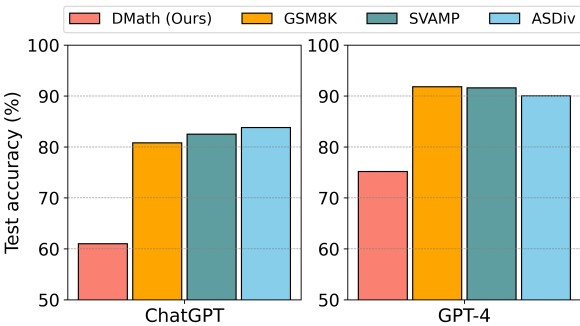

Figure 1: The accuracy of two LLMs on MWP datasets. The result of ChatGPT (gpt-3.5-turbo; OpenAI) and GPT-4 (OpenAI, 2023), using a few-shot CoT (Wei et al., 2022) prompting method, indicate that DMath is the most challenging benchmark. GPT-4 shows about 90% accuracy on three existing datasets and approximately 75% on DMath.

multi-aspect diversity regarding problem types, lexical usage patterns, languages, and intermediate solution forms, as reported in Table 1. The low-diversity dataset leads to over-optimistic results for MWP models by memorizing frequent patterns.

With the advent of large language models (LLMs), they have shown remarkable performances in various reasoning tasks. Notably, the adoption of GPT-4 (OpenAI, 2023) has yielded an impressive 92% accuracy on the GSM8K dataset (Cobbe et al., 2021), which is widely used as a representative MWP benchmark. This result has led to the belief that LLMs demonstrate exceptional proficiency in mathematical reasoning. However, our empirical findings show that the true capability of LLMs has not been revealed because of the limited MWP benchmarks. As illustrated in Figure 1, GPT-4 achieves more than 90% accuracy on the existing three MWP datasets, but its accuracy drops to approximately 75% on our dataset with multi-aspect diversity. This significant gap motivates us to develop more diverse datasets with greater complexity and challenges in real-world problems than existing ones.

---

[1]The dataset is available at https://github.com/JiwooKimAR/dmath

| Dataset | Language | Annotation | Data size | CLD | # Eq. templates | # Total ops |
|---|---|---|---|---|---|---|
| MAWPS (Koncel-Kedziorski et al., 2016) | EN | Expression tree | 1,921 | 0.26 | 39 | 4 |
| MathQA (Amini et al., 2019) | EN | Expression tree | 37,259 | 0.05 | 6,599 | 58 |
| ASDiv-A (Miao et al., 2020) | EN | Expression tree | 1,218 | 0.50 | 19 | 4 |
| SVAMP (Patel et al., 2021) | EN | Expression tree | 1,000 | 0.22 | 26 | 4 |
| GSM8K (Cobbe et al., 2021) | EN | Natural language | 8,792 | 0.81 | - | 4 |
| Math23K (Wang et al., 2017) | ZH | Expression tree | 23,160 | - | 2,187 | 4 |
| Ape210K (Zhao et al., 2020) | ZH | Expression tree | 210,488 | - | 56,532 | 4 |
| **DMath (Ours)** | **EN, KO** | **Expression tree, Python code** | **10,022** | **0.49** | **2,855** | **50** |

Table 1: Statistics of DMath and other MWP datasets. We used corpus lexicon diversity (CLD) proposed in Miao et al. (2020) to quantify the diversity of lexical usage patterns. The higher, the more diverse. The underlined numbers represent the top-3 values in each column. Let EN, KO, and ZH denote English, Korean, and Chinese, respectively, and 'ops' denotes operators.

In this study, we introduce a multi-aspect diverse MWP dataset, called *DMath*, which has the following key features: (1) DMath fully covers problem types across five categories outlined in the math education guideline of grade schools in the USA (CDE, 2006) and Korea (MOE, 2022). These categories include arithmetic (ARI), comparison (COM), correspondence (COR), geometry (GEO), and possibility (POS). Thus, various mathematical reasoning skills can be evaluated on DMath. (2) DMath consists of about 10,000 problems manually created by 43 human workers covering various lexical usage patterns. This diversity in lexical usage patterns helps evaluate the general performance of MWP models. (3) DMath supports bilingual input languages, *i.e.*, English and Korean, to evaluate the effect of input languages for MWP models. (4) We offer the annotation of expression trees and Python code as intermediate solutions to evaluate the effect of expression forms in training MWP models. To annotate the expression trees, we adopt 50 operators, some newly designed to handle the operators for list structures. (Details are given in Appendix B.)

To analyze the performance of various MWP models on our dataset, we employ *fine-tuning* and *prompting* approaches. For the fine-tuning method, we employ three representative MWP models with pre-trained language models, *i.e.*, RoBERTa (Liu et al., 2019), GPT-2 (Radford et al., 2019), and CodeGPT (Lu et al., 2021). For the prompting method, we adopt representative LLMs, *i.e.*, GPT-3 (Brown et al., 2020), ChatGPT (gpt-3.5-turbo; OpenAI), and GPT-4 (OpenAI, 2023), with various reasoning prompting methods, *i.e.*, zero-shot (Brown et al., 2020), zero-shot CoT (Kojima et al., 2022), few-shot CoT (Wei et al., 2022), and PAL (Gao et al., 2022).

Through our empirical studies on DMath, we

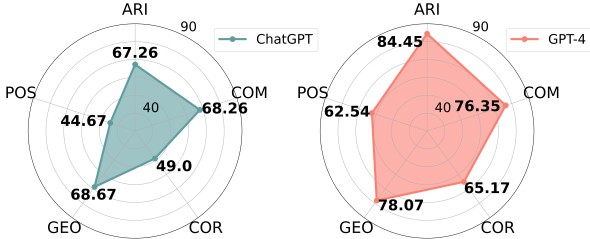

Figure 2: Accuracy comparison over various reasoning categories on DMath for ChatGPT (gpt-3.5-turbo; OpenAI) and GPT-4 (OpenAI, 2023) are presented, respectively. The few-shot CoT (Wei et al., 2022) prompting is used for inference in these models.

found several interesting observations. (1) Due to the multi-aspect diversity, all the MWP models on DMath yield lower performance than other datasets, with ChatGPT and GPT-4 achieving about 60% and 75% accuracy, respectively. (2) The problem types heavily affect the accuracy. As shown in Figure 2, correspondence (COR) and possibility (POS) show a relatively low accuracy, *i.e.*, 44–49% and 62–65% for ChatGPT and GPT-4, respectively. We conjecture that math examples are biased for training LLMs. (3) For different languages, English-based prompting methods perform better than Korean-based ones. As reported in OpenAI (2023), we also confirm that LLMs show more favorable performance on English corpora. (4) For different expression forms, Python code is more effective than other forms. As also observed in Gao et al. (2022), Python code is more robust for addressing logical expressions using external tools.

## 2 Dataset Formulation

The MWP sample comprises a natural narrative (input) and an answer (output). The input includes natural sentences, symbols, and math equations. The output represents a unique numeric or string value. Besides, each sample is associated with its

| | |
|---|---|
| **Question (English)** | You want to pick 2 fruits out of an apple, a peach, a pear and place them in different fruit baskets. What is the number of cases? |
| **Question (Korean)** | 사과, 복숭아, 배 중 2개를 뽑아 서로 다른 과일 바구니에 담으려고 합니다. 가능한 경우의 수는 모두 몇 가지입니까? |
| **Human solution** | len([apple, peach, pear]) permutation 2 |
| **Equation** | [LIST_SOL] apple peach pear [LIST_EOL] [LIST_LEN] 2 [PERM] |
| **Code** | a = 'apple' b = 'peach' c = 'pear' d = 2
li= [] li.append(a) li.append(b) li.append(c)
e = len(li) f = 1
for i, elem in enumerate(range(d)): f = f * (e - i)
print(int(f)) |
| **Answer** | 6 |

Table 2: Example of a math word problem with two languages (English and Korean), two expression forms (an equation and its corresponding Python code), a human solution, and the answer. More examples of DMath for each category can be found in Appendix A.

intermediate solution, *e.g.*, a natural language, an expression tree, or Python code, to derive the answer. Because it is difficult to infer the answer from the narrative directly, the intermediate solution is used as a hint to perform math reasoning and to infer world knowledge from the narrative.

To construct the MWP dataset, we annotate the sample as a triplet of ⟨*natural language narrative*, *intermediate solution*, *answer*⟩. In this process, we focus on extending the diversity of problem types, lexical usage patterns, languages, and intermediate solutions. Table 2 shows a sample with two input languages, *i.e.*, English and Korean, intermediate expression forms for the expression tree and Python code, and the final answer.

**Diversity in problem types.** To cover diverse math concepts, we refer to the math education guidelines of grade schools in the USA (CDE, 2006) and Korea (MOE, 2022). We identify four important reasoning skills in mathematics: *arithmetic*, *relationship*, *geometric*, and *possibility*. Arithmetic reasoning encompasses four fundamental math operations. Relationship reasoning requires the ability to discern relationships between objects in various phenomena. Geometric reasoning entails understanding the nature and relationship of geometric shapes. Possibility reasoning requires the ability to represent uncertainty numerically.

Based on these reasoning tasks, we categorize five problem types, *i.e.*, *arithmetic (ARI)* for arith-

| | ARI | COM | COR | GEO | POS | Total |
|---|---|---|---|---|---|---|
| **Train** | 2,476 | 1,338 | 1,656 | 1,417 | 1,056 | 7,943 |
| **Test** | 669 | 334 | 402 | 383 | 291 | 2,079 |
| **Total** | 3,145 | 1,672 | 2,058 | 1,800 | 1,347 | 10,022 |

Table 3: The number of samples per category on DMath.

metic reasoning, *comparison (COM)* and *correspondence (COR)* for relationship reasoning, *geometry (GEO)* for geometric reasoning, and *possibility (POS)* for possibility reasoning. We split the relationship reasoning into two groups due to their distinct characteristics. Please refer to Appendix A for examples of both categories. Table 3 reports the number of samples for each category.

**Diversity in lexical patterns.** We use corpus lexicon diversity (CLD) introduced by Miao et al. (2020) to quantify lexical usage patterns. Compared to existing MWP datasets, the CLD of our dataset is 0.49. Although it is similar to ASDiv, the number of equation templates in our dataset is 2,855, much greater than ASDiv. It indicates that our problems cover various problem types with high diversity. For detail, a template represents the process of performing an operation. If the order or the type of operators is different, we regard that they are different templates.

**Diversity in language narratives.** Existing datasets (Amini et al., 2019; Wang et al., 2017; Cobbe et al., 2021) are primarily based on monolingual language, *i.e.,* English or Chinese. Our dataset provides bilingual languages, *i.e.*, English and Korean. It is beneficial for validating the effect of languages on LLMs.

**Diversity in solution forms.** To extend the usage, we annotate two expression formats, an expression tree and Python code, as illustrated in Table 2. The math equation has been widely used as an expression tree (Koncel-Kedziorski et al., 2016; Xie and Sun, 2019). Some categories require us to enumerate and select a specific number/string from a list/sequence of numbers/strings. GEO also demands extensive knowledge of geometric shapes like triangles, rectangles, and circles.

To annotate our collected problems, we introduce 50 operators. (See Appendix B.) Following are some rules for defining these operators. First, we create operator names to match actual math symbols as closely as possible. (*e.g.*, [ADD], [FLOOR], [GCD], [LIST_MAX]) Second, when some operators perform the same action, we use similar names. (*e.g.*, (a) [LIST_GET_PERM], (b)

[LIST_GET_PRODUCT], (c) [LIST_GET_DIVISOR]. (a), (b), and (c) all take a LIST as the input and return a LIST as the output through some operation (PERM, PRODUCT, DIVISOR), so "LIST_GET_" overlaps.) Third, we employ concise names, (*e.g.*, [COMB], [LCM], [LIST_LEN], [LIST2NUM].) Lastly, if a single operator performs multiple operations, we represent the goal for operations. (*e.g.*, [DIGIT_UNK_SOLVER]. This operator takes an expression, and the string corresponding to the digits in the expression and finds the unknown numeric value (UNK) of the string.) Given a postfix expression tree, we also convert it into Python code automatically using our Python code generator. As the byproduct of our code generator, we can thus utilize Python code as another solution form.

## 3 Dataset Construction

While other MWP datasets are typically collected from the web and annotated with relevant equations, the DMath dataset exclusively comprises human-generated problems. The linguistic diversity is higher than others regarding natural languages and expression trees. Specifically, we provide a comprehensive overview of collecting, augmenting, annotating, validating, and translating the DMath dataset.

**Collection.** Instead of curating existing math problems, we ask human workers to create new problems. We recruit 43 undergraduate and graduate students and let them develop new and diverse grade school-level math problems. Each worker contributes to creating about 100–200 samples, resulting in a total collection of 4,184 seed samples. These samples were distributed across five math categories, with 960 for arithmetic, 1,092 for comparison, 943 for correspondence, 401 for geometry, and 788 for possibility.

**Augmentation.** Human workers manually augment seed samples using three policies: paraphrasing, domain word changes, and expanding equation trees (using a few more equations from the original sentence problem or adding other equations). We carefully augment seed samples to minimize linguistic similarities and superficial differences. Human workers manually boost 1–3 samples from each seed sample using different expressions. We also verify the similarity between seed and augmented samples using BLEU (Papineni et al., 2002) scores and remove some samples with high similarities. The statistics for BLEU score between seeds and

augmented problems are 0.0 for min, 0.24 for mean, 0.23 for median, and 0.92 for max. Refer to the detailed histogram in Appendix C.

**Annotation.** We request the worker to annotate its answer and the intermediate solution as an expression tree using pre-defined operators. This expression tree follows the postfix notation and consists of numbers, strings, and operators, as exemplified in Table 2. (See Appendix A for more examples.) While existing studies (Koncel-Kedziorski et al., 2015; Wang et al., 2017; Cobbe et al., 2021) solely focus on variables for single numbers, we introduce new variables and operators for list structure, which can be used to solve complex and various math problems. We implement an automatic code generator to convert the expression tree into Python code. We also use the Python code to validate whether the human-annotated answer is correct.

**Validation.** We recruit 13 graduate students majoring in computer science who participate in the previous process (*collection, augmentation* and *annotation*). They are familiar with the annotation format of the MWP dataset and can solve elementary-level mathematical problems well. We perform the following process sequentially: (1) Question-answer validation, (2) Question-expression tree validation, (3) Expression tree validation, and (4) Conciseness validation of expression tree for multi-expression tree matching problems. If any error was detected during steps (1) to (3), we return to step (1) for re-validation. After that, we perform step (4).

We go through these procedures to reduce potential risks to the correctness of the annotated data. For step (1) and step (2), we ensure that the questions assigned to human workers do not overlap so that they do not receive the same questions in each step, which can act as a cross-check. We employ code to check step (3). For step (4), 2-3 human workers are assigned per question. More detailed procedures are as follows.

**Step (1): Question-answer validation.** After the *annotation* process, we assign natural language questions to human workers and ask them to solve the provided questions. There are two cases where their answers differ from the annotated answers. The first is when the natural language problem is unsolvable, and the second is when the annotated answer is incorrect. When a worker encounters these errors while solving a problem, the worker corrects the error manually. In the first case, the

worker should correct the natural language question. In the second case, the worker should correct the annotated answer. Besides, two or more workers cross-check the corrected questions to ensure no errors.

**Step (2): Question-expression tree validation.** Natural language questions that are solvable and correctly annotated with correct answers are given to human workers along with expression trees. The human workers check whether the given natural language question and the expression tree are correctly paired. If not, they modify the expression tree to match the natural language problem.

**Step (3): Expression tree validation.** After the natural language question and the expression tree have been properly matched in the previous step, we verify that the expression tree produces the correct answer. The operators used in expression trees can be converted to Python code. After converting each expression tree to code, we verify if the outcome produced upon execution corresponds to the correct answer. Otherwise, return to step (1).

**Step (4): Conciseness validation of expression tree for multi-expression tree matching problems.** There can be various potential solutions to a mathematical problem, and we maintain that selecting an expression tree is crucial to the model training. For example, the following problem has multiple expression trees. *"There are six people. Everyone shook hands with each other once. Find the total number of handshakes."* We can make the expression tree as one of `6*(6-1)/2` or `6C2`. Here, both equations lead to a reasonable correct answer. We choose the expression tree with as short an equation length as possible. (For this example, we choose `6C2`.) We considered this direction of reducing the equation length because the most concise solution process captures the essence of the problem. However, even if a formula is not the shortest, we choose it if most workers agree.

**Translation.** Once we build the Korean MWP dataset, we translate it into English. First, we use machine translation to translate Korean into English to improve the quality and quantity of translations in a limited time using Google Translator. Then, we ask nine English-majored students to revise any incorrect translations. This revision ensures that the English and Korean problems share the same meaning. It can also improve the linguistic diversity of the English language. For the whole data (# = 10,022), translators change 71% of prob-

lems (# = 7,121) from machine-translated ones. We further inspect 29% problems (# = 2,901) to ensure no semantic/grammatical errors. Afterward, all questions (# = 10,022) are checked for those errors. This is done by three graduate students who are not involved in the translation and are proficient in English and Korean. The results show that 5.24% (# = 525) of the questions had semantic/grammatical errors.

# 4 Experimental Setup

**Datasets.** We use three datasets to compare with DMath, which are GSM8K (Cobbe et al., 2021), SVAMP (Patel et al., 2021), and ASDiv (Miao et al., 2020). GSM8K is a commonly used dataset for evaluating math reasoning and has high quality. SVAMP is a variation of the previous dataset (MAWPS) (Koncel-Kedziorski et al., 2016), and it allows us to evaluate the robustness of the methods. ASDiv is a highly diverse dataset, showing the importance of increasing the lexical diversity of the problem. We did not run experiments on the MathQA (Amini et al., 2019) dataset because we found that it has incorrect answers, as also discussed in existing studies (Austin et al., 2021; Jie et al., 2022; Miao et al., 2020). Also, we did not compare DMath with MATH (Hendrycks et al., 2021) since there is a discrepancy in domain knowledge levels. We aimed to tackle math problems at the elementary school level (grades 1-6), while MATH is designed for the high school level (grades 8-12). See more details in Appendix D.

**Baselines.** We evaluate the MWP solvers with the fine-tuning method and the prompting method. For the fine-tuning method, we use pre-trained language models. We use two representative models, RoBERTa (Liu et al., 2019) and GPT-2 (Radford et al., 2019) for the expression tree output and use CodeGPT (Lu et al., 2021) for the Python code output. They are fine-tuned with DMath train data. For more information on how the framework works, refer to Appendix E.

For the prompting method, we use GPT-based models. We use babbage, curie, davinci (Brown et al., 2020), gpt-3.5-turbo (OpenAI), and gpt-4 (OpenAI, 2023) model using API provided by OpenAI. We denote babbage as GPT-3 [1B], curie as GPT-3 [6.7B], davinci as GPT-3 [175B], gpt-3.5-turbo as ChatGPT [175B], and gpt-4 as GPT-4 [Unknown]. Besides, we use four prompting methods: zero-shot (Brown et al., 2020), zero-shot

| Framework | # Params. | Model | Prompt method | ARI | COM | COR | GEO | POS | Total |
|-----------|-----------|-------|---------------|-----|-----|-----|-----|-----|-------|
| Fine-tuning | 124M | GPT-2$_{BASE}$ | - | 39.46 | 44.31 | 41.46 | 20.45 | 38.03 | 36.92 |
| | 125M | RoBERTa$_{BASE}$ | - | 41.60 | 36.92 | 42.37 | 26.28 | 29.90 | 36.54 |
| | 355M | RoBERTa$_{LARGE}$ | - | 45.99 | 42.12 | 49.01 | 32.03 | 35.05 | 41.85 |
| | 774M | GPT-2$_{LARGE}$ | - | 50.67 | 51.10 | 57.30 | 31.15 | 44.44 | 47.56 |
| Prompting | 1B | GPT-3 | Few-shot CoT | 4.63 | 25.75 | 4.98 | 6.53 | 8.25 | 8.95 |
| | 6.7B | | | 6.58 | 20.66 | 3.48 | 4.44 | 8.93 | 8.18 |
| | 175B | | | 18.68 | 37.13 | 9.20 | 10.44 | 10.31 | 17.12 |
| | 175B | ChatGPT | Zero-shot | 47.68 | 38.32 | 23.13 | 37.08 | 29.90 | 36.99 |
| | | | Zero-shot CoT | 64.87 | 69.76 | 45.77 | 64.23 | 36.77 | 57.91 |
| | | | Few-shot CoT | 67.26 | 68.26 | 49.00 | 68.67 | 44.67 | 60.99 |
| | Unknown | GPT-4 | Zero-shot | 49.93 | 50.00 | 28.11 | 46.21 | 35.74 | 43.05 |
| | | | Zero-shot CoT | 85.65 | 87.43 | 65.92 | 81.72 | 62.54 | 78.16 |
| | | | Few-shot CoT | 84.45 | 76.35 | 65.17 | 78.07 | 62.54 | 75.18 |

Table 4: Accuracy comparison results of MWP models using expression tree solution form, GPT-2 (Radford et al., 2019), RoBERTa (Liu et al., 2019), GPT-3 (Brown et al., 2020), ChatGPT (OpenAI), and GPT-4 (OpenAI, 2023), on the English-based DMath dataset. Due to space limitation, we omit zero-shot (Brown et al., 2020) and zero-shot CoT (Kojima et al., 2022) results for GPT-3, which follow a similar trend to that of few-shot CoT.

CoT (Kojima et al., 2022), few-shot CoT (Wei et al., 2022), and PAL (Gao et al., 2022) to compare the effect of different prompting methods.

**Evaluation metrics.** To evaluate MWP solvers, we use *accuracy (Acc.)*, commonly used in existing studies. The accuracy checks whether the final answer is correct without considering intermediate solution forms. To extract the answers from the MWP model, we use the following methods. For the fine-tuning method, we convert the expression tree solution form generated by models to Python code and take the result of the code as the answer. A model that generates Python code as the intermediate output also takes the result of the code as the answer. For the prompting method, we adopt the majority voting method. It selects the most frequent answer among the possible answers as the final answer. We treat any answer after *"answer is"* as a possible answer.

**Implementation details.** For the fine-tuning models, we use pre-trained models from the hugging-face hub. We set the batch size to 32 and the maximum token length to 50. We fine-tune models for three learning rates {1e–5, 3e–5, 5e–5} to find the best performance. We also run experiments with three different seed values for each learning rate, then select the best-performing learning rate and report the average value over the three seeds. We run our experiments using an NVIDIA TITAN-RTX GPU with 24GB memory.

For prompting-based methods, we set the temperature as 1.0 and top_p as 1.0. We also set the number of few-shot examples as 8 for

GPT-3 (babbage, curie, davinci) and ChatGPT (gpt-3.5-turbo-0301), and set 5 for GPT-4 (gpt-4-0314) following the conventions (Wei et al., 2022; OpenAI, 2023). For GPT-3, we set the maximum token length to 128. Also, we set few-shot examples of GSM8K, SVAMP, and ASDiv as Wei et al. (2022) and of DMath as Appendix F. We take them differently for a fair comparison.

## 5 Empirical Results

**Overall performance results.** Table 4 presents the overall accuracies of fine-tuning and prompting MWP models. First, owing to the impressive capabilities of LLMs, GPT-4 with few-shot CoT shows the highest accuracy, achieving 75.18%. For GPT-3 [175B] and ChatGPT with the same number of parameters, there exists a notable performance gap. We conjecture that ChatGPT has undergone more optimization for mathematical reasoning compared to GPT-3. It is also worth noting that all fine-tuning models surpass GPT-3, and even the fine-tuned GPT-2$_{LARGE}$ [774M] exhibits comparable accuracy to GPT-4 with zero-shot. This suggests that the fine-tuning approach proves effective in optimizing the MWP task.

Second, the worst problem categories differ across MWP models. Both GEO and POS rely on world knowledge, impacting the performance of fine-tuning models, *i.e.*, RoBERTa$_{LARGE}$ and GPT-2$_{LARGE}$, differently. Fine-tuned models excel at POS due to condensed operations but struggle with expression tree-based problems in GEO. Conversely, prompting models, *i.e.*, ChatGPT and GPT-

| Expression form | Model | # Params. | Prompt method | ARI | COM | COR | GEO | POS | Total |
|---|---|---|---|---|---|---|---|---|---|
| Expression tree | RoBERTa$_{BASE}$ | 125M | - | 41.60 | 36.92 | 42.37 | 26.28 | 29.90 | 36.54 |
| Python code | CodeGPT$_{SMALL}$ | 124M | - | 42.25 | 35.83 | 32.75 | 23.24 | 33.34 | 34.63 |
| NL prompt | ChatGPT | 175B | Few-shot CoT | 67.26 | 68.26 | 49.00 | 68.67 | 44.67 | 60.99 |
| Python code prompt | | | PAL | 74.74 | 52.99 | 52.49 | 61.62 | 46.39 | 60.56 |
| NL prompt | GPT-4 | Unknown | Few-shot CoT | 84.45 | 76.35 | 65.17 | 78.07 | 62.54 | 75.18 |
| Python code prompt | | | PAL | 88.94 | 83.83 | 69.40 | 81.72 | 67.35 | 79.99 |

Table 5: Accuracy comparison results of MWP models on the DMath dataset per expression forms in English. NL prompts used by ChatGPT (OpenAI) and GPT-4 (OpenAI, 2023) mean natural language prompts. We set few-shot CoT (Wei et al., 2022) as NL prompts method and PAL (Gao et al., 2022) as Python code prompt. We choose RoBERTa$_{BASE}$ (Liu et al., 2019) as the comparison model for CodeGPT$_{SMALL}$ (Lu et al., 2021) because it has the most similar parameter sizes.

4 with few-shot CoT, trained on extensive world knowledge, perform better in GEO but falter with the Number of Cases problems in POS. In COR, the models' difficulty understanding specific problem formats contributes to performance degradation.

Third, the prompting methods in LLMs also incur distinct results. ChatGPT and GPT-4 show the lowest performance for the zero-shot prompting. They can be significantly improved by using CoT prompting. However, when model sizes are too small, the impact of few-shot CoT prompting is insignificant, implying that model capacity is an important factor in prompting-based models.

Lastly, we summarize our observations on five problem types. ARI and COM can be regarded as common math types frequently encountered in natural language corpora. These properties explain the relatively higher accuracies of ARI and COM than other types in both fine-tuning and prompting methods. Meanwhile, COR is a less frequent problem type in ordinary language corpora, but an appropriate substitution can solve it. This unique characteristic of COR lends itself to the clear advantage of fine-tuning approach and CoT. GEO often requires world knowledge, thus, the prompting method with LLMs is more advantageous than the fine-tuning methods. Finally, POS is represented by relatively longer equation trees, indicating a challenging problem type in general. The operations and mathematical expressions for POS are also domain-specific, limiting the generalization power of LLMs and hindering their performance in this problem type.

**Expression form results.** Table 5 presents the result of different expression formats. Firstly, ChatGPT achieves 60.99% with natural language prompts and 60.56% with Python code prompts. While the performance of the models using natural language prompts and those using Python code

prompts appear similar, a closer look reveals that a specific prompt type is preferred depending on the problem. Among the 2,079 questions in the full test set, 15.39% (# = 320) of the questions are correctly solved only on the natural language prompt, 45.60% (# = 948) are correctly solved on both the natural language prompt and the Python code prompt, 14.96% (# = 311) are correctly solved on only the Python code prompt, and 24.05% (# = 500) were incorrect.

We further analyze these results by problem type. For ARI types, 8.97% of problems are correct using only natural language prompts and 16.44% are correct using only Python code prompts. For COM, the corresponding percentages are 29.34% and 14.07%, respectively. ARI and COM show the largest performance differences among the five categories between natural language prompt and Python code prompt. Python code prompts perform better on ARI types because LLM often needs correction in its calculations. The higher accuracy of natural language prompts in COM types can be attributed to the concise language expression of large and small relationships between two objects, compared to Python code.

Lastly, the fine-tuning and prompting methods show similar tendencies in model performance. This tendency indicates that various forms of expression have their preferred categories. To elaborate, when we state that form of expression A outperforms form B, it does not imply that every type within A uniformly surpasses every type within B. Rather, it suggests that certain types within A outperform B, while some types in B excel over A. Within the fine-tuning framework, we compare the RoBERTa$_{BASE}$ with the CodeGPT$_{SMALL}$. The RoBERTa$_{BASE}$ model uses an expression tree as the solution form, whereas the CodeGPT$_{SMALL}$ model uses Python code. Here,

| Dataset | Model | Prompt method | |
| --- | --- | --- | --- |
| | | **Zero-shot CoT** | **Few-shot CoT** |
| GSM8K | ChatGPT | 78.85 | 80.82 |
| | GPT-4 | 93.25 | 91.81 |
| SVAMP | ChatGPT | 77.70 | 82.50 |
| | GPT-4 | 88.10 | 91.60 |
| ASDiv | ChatGPT | 79.71 | 83.80 |
| | GPT-4 | 88.55 | 90.03 |
| DMath | ChatGPT | 57.91 | 60.99 |
| | GPT-4 | 78.16 | 75.18 |

Table 6: Accuracy comparison between DMath and GSM8K (Cobbe et al., 2021), SVAMP (Patel et al., 2021), ASDiv (Miao et al., 2020) for different prompt methods in English.

the length of the output sequence emerges as the most significant impact. In detail, for ARI and POS types, CodeGPT$_{SMALL}$ using Python code performs better, and for COM, COR, and GEO types, RoBERTa$_{BASE}$ model using expression tree performs better. The average token length and std. for COR types with expression tree are (6.61, 4.13), which is lower than the POS type of (10.81, 4.51). That of POS type with Python code is (129.54, 57.55), which is lower than COR type of (147.61, 205.04). In common, GEO types seem to have lower token lengths but lower performance due to the lack of domain knowledge of pre-trained models.

**Comparison on different datasets.** Table 6 shows the results of two different prompting methods on four datasets, including DMath. Except for DMath, the GSM8K, SVAMP, and ASDiv datasets, all achieve the accuracy higher than 90% on GPT-4 and surpass 80% on ChatGPT in few-shot prompts. (Figure 1 showcases the few-shot prompt results per dataset.) The zero-shot CoT results show a similar trend. This trend can be attributed to the fact that these datasets primarily handle arithmetic reasoning problems.

Instead of intentionally inflating the difficulty to undermine model performance, we have noted significant performance differences between models on our dataset. Specifically, DMath exhibits average deviations of 13.07% and 15.35% in zero-shot CoT and few-shot CoT prompting methods, respectively, on the GPT-4 model. This suggests that our dataset contains problems that can be incrementally solved as model capacity improves. The problems are challenging yet reasonable and tailored to grade school levels. In brief, our dataset pinpoints areas

| Few-shot CoT Prompt | Acc. (%) |
| --- | --- |
| Few-shot examples of (Wei et al., 2022) | 57.96 |
| Uniform category random | 58.97 |
| Reverse uniform category random | 58.68 |
| Random | 59.31 |
| One category random | 60.99 |

Table 7: Accuracy for several few-shot CoT (Wei et al., 2022) prompting methods on the English-based DMath dataset.

where large language models falter, setting a pathway for future enhancements.

**Few-shot selection.** The few-shot prompt is given DMath and other datasets (GSM8K, SVAMP, and ASDiv) differently. Other datasets used the example of Wei et al. (2022) as in previous studies. Unlike them, DMath is a mix of different mathematical categories. We consider evaluating DMath with the same prompt as other arithmetic mathematical reasoning datasets to be unfair because there are several problem types in DMath. If we give few-shot examples of only arithmetic type, then the model will not do well except arithmetic. Therefore, various prompt selection methods were taken, and the highest-performance prompt method, *one category random* method was used as the final few-shot prompt examples in this paper. (See more details in Appendix F.)

We validate the best-performing prompt method by comparing five few-shot CoT prompts. Table 7 shows the performance for each prompt method. First, we use the few-shot example of Wei et al. (2022) as a baseline. The other few-shot CoT prompt selects examples from DMath train data. The *uniform category random* selects examples according to the proportion of problems in each category; it selects 2 examples from ARI, COR, and GEO and 1 example from COM and POS each. The *reverse uniform category random* selects 1 problem from ARI, COR, GEO and 2 problems from COM, POS each. The *Random* selects examples randomly from the whole dataset. Finally, the *one category random* selects few-shot examples based on the type of question, *i.e.*, if the model gets an ARI type question, it randomly selects few-shot examples of ARI type, and if it gets a POS type question, it selects few-shot examples of POS type.

**Language-specific performance of LLMs.** Table 8 provides the results of different input languages, English and Korean. By using ChatGPT and GPT-4, we observe that the English-based prompt method performs better than its Korean-

| Model | Prompt method | Language | |
|---|---|---|---|
| | | EN | KO |
| ChatGPT | Few-shot CoT | 60.99 | 46.03 |
| GPT-4 | | 75.18 | 65.37 |

Table 8: Accuracy comparison results of MWP models, ChatGPT (OpenAI) and GPT-4 (OpenAI, 2023), on DMath. Few-shot CoT (Wei et al., 2022) is used as the prompting method.

based counterparts despite both English and Korean problems being identical in content but expressed in different languages. Ideally, if the language understanding is independent of the mathematical reasoning, it should have produced the same accuracy. However, accuracy has shown more than a 13% and 24% difference between the two on GPT-4 model and ChatGPT, respectively. We conjecture that the data imbalance issue of LLMs leads to performance bias, as pointed out in several papers (Blodgett et al., 2020; Lai et al., 2023).

**Qualitative analysis of GPT-4 failure cases.** We conduct a qualitative analysis on failure cases of GPT-4 with few-shot CoT. (See several examples in Appendix G.) For ARI, COM, and GEO examples, as the length of the reasoning path increases, GPT-4 generates incorrect answers more frequently. For COR, the problems about the number of digits in a number mostly incur incorrect answers. For GEO, GPT-4 failed to elicit certain domain knowledge, *e.g.*, the rectangle that can be made to have the largest area given a perimeter is a square. For POS, we observe GPT-4 found difficulties in solving problems related to the Number of Cases.

## 6 Related Work

**Math word problem solving datasets.** Automatic MWP-solving has been explored since the 1960s (Bobrow, 1964), with work primarily focused on *corpus collection* and *model development*. Initially, small datasets were used (Kushman et al., 2014; Upadhyay and Chang, 2015; Koncel-Kedziorski et al., 2015; Hosseini et al., 2014; Roy et al., 2015; Roy and Roth, 2017; Seo et al., 2014; Shi et al., 2015). The MAWPS dataset (Koncel-Kedziorski et al., 2016) unified these small-scale datasets and incorporated various problem types. Following studies like (Huang et al., 2016; Ling et al., 2017; Amini et al., 2019; Zhao et al., 2020) enriched this further, but they were limited by their reliance on multiple choice and a lack of diverse lexical usage patterns. Recent works (Cobbe et al.,

2021; Hendrycks et al., 2021) targeted a broader diversity, but more enhancement is needed. Our dataset addresses this by providing diversity in problem types, lexical usage patterns, languages, and expression forms.

**Math word problem solving models.** Early MWP models used symbolic approaches (Kintsch and Greeno, 1985; Fletcher, 1985), but their domain specificity limited their use. Later, template-based and statistical methods (Kushman et al., 2014; Hosseini et al., 2014; Mitra and Baral, 2016; Roy and Roth, 2018; Koncel-Kedziorski et al., 2015) improved performance but faced challenges with diverse datasets (Huang et al., 2016). Deep neural networks became popular, with RNN-based models (Wang et al., 2017, 2019), tree generation models (Xie and Sun, 2019; Zhang et al., 2020; Wu et al., 2021), and relation extraction models (Jie et al., 2022). Recent models leverage pre-trained language models like BERT (Devlin et al., 2019) to enhance performance (Tan et al., 2021; Li et al., 2021; Kim et al., 2020). Large language models have also shown significant performance improvements with techniques like step-by-step reasoning (Wei et al., 2022; Nye et al., 2021), calculation verifiers (Cobbe et al., 2021), and various prompting methods (Kojima et al., 2022; Wei et al., 2022; Gao et al., 2022). Recent studies (Huang et al., 2022; Wang et al., 2022; Madaan et al., 2023) indicate that large language models can improve reasoning capabilities autonomously without supervised data, surpassing current techniques in accuracy.

## 7 Conclusion

This paper proposed DMath, a collection of 10K high-quality grade school-level math word problems. It mainly pursued multi-aspect diversity for problem types, lexical usage patterns, languages, and expression forms. We evaluated various MWP models on DMath and other datasets. We observed that the logical reasoning capability of the MWP model was mainly focused on arithmetic reasoning. Because of various features, our dataset is more beneficial for inspecting the diverse reasoning abilities of models. Our dataset can also help evaluate MWP models for diverse aspects.

## Limitations

There are several limitations to this work. (1) Translating and annotating math problems require high human costs. Although we carefully and manually

check all the samples and their annotations, our dataset might include a few samples with incorrect answers. (2) Although it is interesting to observe various model scales, evaluating pre-trained language models with different model sizes requires GPU resources and training time. Therefore, we employ three well-known pre-trained models for evaluation.

## Ethics Statement

Following the ACL Code of Ethics, we emphasize the ethical facets of data collection, use, and the potential ramifications of our research within artificial intelligence. This study presents a new mathematical reasoning dataset, and we have carefully examined and addressed all ethical considerations tied to its creation and usage.

The construction of this mathematical reasoning dataset involved authors and human workers in the data collection, creation, and augmentation phases. All contributors have strictly followed ethical guidelines, ensuring a respectful and fair environment. We have prioritized maintaining the privacy and anonymity of all parties involved.

We have gone to great lengths to confirm that the data gathered does not pose any risk of harm to grade school students, the primary demographic of our study. The content has been conscientiously curated to exclude harmful or inappropriate material, fostering a safe learning environment.

The dataset's accuracy is of utmost importance, and every attempt has been made to ensure the validity of the labels. Human reviewers have manually checked all answer labels in the dataset. However, despite our rigorous efforts, we acknowledge that some errors may still exist due to the inherent limitations of manual review and the complexity of the task. We encourage users of the dataset to remain mindful of this limitation and to report any errors they encounter for continuous improvement of the dataset.

We also recognize the potential misuse of the dataset, and we ardently advocate for its ethical and responsible usage within the AI community. This dataset aims to advance the understanding of mathematical reasoning among grade school students, and any use that could compromise fairness or propagate biases in AI systems is strongly discouraged.

In conclusion, we strive to ensure our dataset contributes to the field of AI and mathematical rea-

soning in a manner that aligns with the ACL's Code of Ethics. We encourage ongoing scrutiny and discussion around the dataset's ethical implications within the research community. We are committed to balancing research innovation with ethical considerations and stand ready to address any concerns raised by the ethics committee or users of the dataset.

## Acknowledgments

This work was supported by Institute of Information & communications Technology Planning & Evaluation (IITP) grant funded by the Korea government (MSIT) (No. 2019-0-00421, No.2022-0-00006, No. 2022-0-00680, and RS-2023-00219919).

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

## Table of Contents

## A   More examples of DMath

---

### CATEGORY: ARITHMETIC CALCULATION

**Question in English**

There are three numbers 10, 11 and 12. What is the remainder when the smallest number is divided by the second smallest number?

**Question in Korean**

3개의 수 10, 11, 12가 있습니다. 그 중에서 가장 작은 수와 2번째로 작은 수로 나눈 나머지는 얼마입니까?

**Equation**

[LIST_SOL] 10 11 12 [LIST_EOL] 1 [LIST_MIN] 2 [LIST_MIN] [MOD]

**Code**

```
var_a = 10\n    var_b = 11\n    var_c = 12\n    list_a= []\n

if "/" in str(var_c):\n    var_c = eval(str(var_c))\n    list_a.append(var_c)\n

if "/" in str(var_b):\n    var_b = eval(str(var_b))\n    list_a.append(var_b)\n

if "/" in str(var_a):\n    var_a = eval(str(var_a))\n    list_a.append(var_a)\n

list_a.reverse()\n    var_d = 1\n    list_b=list_a.copy()\n    list_b.sort()\n

var_e = list_b[var_d-1]\n    var_f = 2\n    list_c=list_a.copy()\n    list_c.sort()\n

var_g = list_c[var_f-1]\n    var_h = var_e % var_g\n

print(int(var_h))
```

**Answer**

**Question in English**

A is 49 plus 33, B is 9 multiplied by 9. Which of the two numbers A and B is greater?

**Question in Korean**

A는 49에서 33을 더한 수, B는 9에서 9를 곱한 수이다. A, B 두 수 중 더 큰 수는 무엇인가?

**Equation**

[LIST_SOL] A B [LIST_EOL] [LIST_SOL] 49 33 [ADD] 9 9 [MUL] [LIST_EOL] 1 [LIST_MAX] [LIST_INDEX] [LIST_POP] [LIST_GET]

**Code**

```
var_a = 'A'\n    var_b = 'B'\n    list_a= []\n

if "/" in str(var_b):\n    var_b = eval(str(var_b))\n    list_a.append(var_b)\n

if "/" in str(var_a):\n    var_a = eval(str(var_a))\n    list_a.append(var_a)\n

list_a.reverse()\n    var_c = 49\n    var_d = 33\n    var_e = var_c + var_d\n

var_f = 9\n    var_g = 9\n    var_h = var_f * var_g\n    list_b= []\n

if "/" in str(var_h):\n    var_h = eval(str(var_h))\n    list_b.append(var_h)\n

if "/" in str(var_e):\n    var_e = eval(str(var_e))\n    list_b.append(var_e)\n

list_b.reverse()\n    var_i = 1\n    list_c=list_b.copy()\n    list_c.sort()\n

var_j = list_c[-var_i]\n    var_k = list_b.index(var_j)+1\n    var_l = list_a[var_k-1]

print(var_l)
```

**Answer**

A

**Question in English**

If the six-digit number 341ABC is divisible by 7, 9, and 13, how much is the three-digit number ABC?

**Question in Korean**

여섯 자리 수 341ABC이 7, 9, 13으로 나누어 떨어질 때, 세 자리 수 ABC은 얼마인지 구하여라.

**Equation**

341ABC [GEN_POSSIBLE_LIST] 7 [LIST_DIVISIBLE] 9 [LIST_DIVISIBLE] 13 [LIST_DIVISIBLE] 341ABC A [LIST_FIND_UNK] 100 [MUL] 341ABC B [LIST_FIND_UNK] 10 [MUL] 341ABC C [LIST_FIND_UNK] [ADD] [ADD]

**Code**

var_a = '341ABC'\n    ans_dict = dict()\n    var_a = str(var_a)\n    list_a = []\n

variable_candi = set(['A', 'B', 'C', 'D', 'E', 'F', 'G', 'H', 'I', 'J', 'K', 'L', 'M', 'N', 'O', 'P', 'Q', 'R', 'S', 'T', 'U', 'V', 'W', 'X', 'Y', 'Z'])\n

for v in set(var_a):\n    if v in variable_candi:\n    ans_dict[v] = 0\n

candi = list(itertools.product('0123456789', repeat=len(ans_dict)))\n

for c in candi:\n    temp = var_a\n    for i, (k, _) in enumerate(ans_dict.items()):\n    temp = temp.replace(k, str(c[i]))\n

if len(var_a) == len(str(int(temp))):\n    new_elem = int(temp)\n    list_a.append(new_elem)\n

var_b = 7\n    list_b = []\n    var_b = int(var_b)\n    for i in list_a:\n    i = int(i)\n    if i % var_b == 0:\n list_b.append(i)\n    var_c = 9\n    list_c = []\n

var_c = int(var_c)\n    for i in list_b:\n    i = int(i)\n    if i % var_c == 0:\n    list_c.append(i)\n

var_d = 13\n    list_d = []\n    var_d = int(var_d)\n    for i in list_c:\n    i = int(i)\n

if i % var_d == 0:\n    list_d.append(i)\n    var_e = '341ABC'\n    var_f = 'A'\n    var_e = str(var_e)\n

var_f = str(var_f)\n    unk_idx = var_e.index(var_f)\n    var_g = 0\n    for elem in list_d:\n

elem = str(elem)\n    var_g = int(elem[unk_idx])\n    var_h = 100\n    var_i = var_g * var_h\n

var_j = '341ABC'\n    var_k = 'B'\n    var_j = str(var_j)\n    var_k = str(var_k)\n

unk_idx = var_j.index(var_k)\n    var_l = 0\n    for elem in list_d:\n    elem = str(elem)\n

var_l = int(elem[unk_idx])\n    var_m = 10\n    var_n = var_l * var_m\n    var_o = '341ABC'\n

var_p = 'C'\n    var_o = str(var_o)\n    var_p = str(var_p)\n    unk_idx = var_o.index(var_p)\n

var_q = 0\n    for elem in list_d:\n    elem = str(elem)\n    var_q = int(elem[unk_idx])\n

var_r = var_n + var_q\n    var_s = var_i + var_r\n

print(int(var_s))

**Answer**

**Question in English**

We are trying to stick a tape in a straight line. The three tapes cut are 36 centimeters (cm), 42 centimeters (cm), and 48 centimeters (cm), respectively, and if the same part is overlapped and attached, the total length is 97 centimeters (cm). Find the length of the overlapping part.

**Question in Korean**

테이프를 일직선으로 쭉 붙이려고 한다. 잘라낸 3개의 테이프가 각각 36센티미터(㎝), 42센티미터(㎝), 48센티미터(㎝)이고 같은 부분만큼 겹쳐서 붙이면 총 길이가 97센티미터(㎝)이다. 겹쳐진 부분의 길이는 얼마인지 구하시오.

**Equation**

36 42 [ADD] 48 [ADD] 97 [SUB] 3 1 [SUB] [DIV]

**Code**

```
var_a = 36\n    var_b = 42\n    var_c = var_a + var_b\n    var_d = 48\n

var_e = var_c + var_d\n    var_f = 97\n    var_g = var_e - var_f\n

var_h = 3\n    var_i = 1\n    var_j = var_h - var_i\n    var_k = var_g / var_j\n

print(':.2f'.format(round(var_k+1e-10,2)))
```

**Answer**

14.50

**Question in English**

You want to form a four-digit number using the four numbers 7, 2, 5, and 9 only once. How many four-digit numbers can you make? **Question in Korean**

4개의 숫자 7, 2, 5, 9를 한 번씩만 사용하여 네 자리 수를 만들려고 합니다. 만들 수 있는 네 자리 수는 모두 몇 개입니까?

**Equation**

[LIST_SOL] 7 2 5 9 [LIST_EOL] 4 [LIST_GET_PERM] [LIST_LEN]

**Code**

```
print(int(var_af))
var_a = 7\n    var_b = 2\n    var_c = 5\n    var_d = 9\n    list_a= []\n
if "/" in str(var_d):\n    var_d = eval(str(var_d))\n    list_a.append(var_d)\n
if "/" in str(var_c):\n    var_c = eval(str(var_c))\n    list_a.append(var_c)\n
if "/" in str(var_b):\n    var_b = eval(str(var_b))\n    list_a.append(var_b)\n
if "/" in str(var_a):\n    var_a = eval(str(var_a))\n    list_a.append(var_a)\n
list_a.reverse()\n    var_e = 4\n    list_b = [str(i) for i in list_a]\n
list_b = list(itertools.permutations(list_b, var_e))\n    list_b = [''.join(num_list) for num_list in list_b]\n
list_b = [str_num for str_num in list_b if str_num[0] != '0']\n    list_b = [float(i) for i in list_b]\n
var_f = len(list_b)\n
print(int(var_f))
```

**Answer**

# B Operators used in expression tree solutions

$\mathcal{R}$: real number, $\mathcal{N}$: Natural number, $\mathcal{L}$: List, $\mathcal{S}$: String, $\mathcal{V}$: Variable, $\mathcal{E}$: Equation

| Operation | Explanation | Examples |
|---|---|---|
| [ADD] | Get $\mathcal{R}_1, \mathcal{R}_2 \Rightarrow$ Return $\mathcal{R}_1 + \mathcal{R}_2$ | 1 2 [ADD] $\rightarrow$ 3 |
| [SUB] | Get $\mathcal{R}_1, \mathcal{R}_2 \Rightarrow$ Return $\mathcal{R}_1 - \mathcal{R}_2$ | 2 1 [SUB] $\rightarrow$ 1 |
| [DIV] | Get $\mathcal{R}_1, \mathcal{R}_2 \Rightarrow$ Return $\mathcal{R}_1 / \mathcal{R}_2$ | 6 2 [DIV] $\rightarrow$ 3 |
| [MUL] | Get $\mathcal{R}_1, \mathcal{R}_2 \Rightarrow$ Return $\mathcal{R}_1 * \mathcal{R}_2$ | 7 5 [MUL] $\rightarrow$ 35 |
| [FDIV] | Get $\mathcal{R}_1, \mathcal{R}_2 \Rightarrow$ Return $\mathcal{R}_1 // \mathcal{R}_2$ | 5 3 [FDIV] $\rightarrow$ 1 |
| [MOD] | Get $\mathcal{R}_1, \mathcal{R}_2 \Rightarrow$ Return $\mathcal{R}_1 \% \mathcal{R}_2$ | 5 3 [MOD] $\rightarrow$ 2 |
| [POW] | Get $\mathcal{R}_1, \mathcal{R}_2 \Rightarrow$ Return $\mathcal{R}_1 \wedge \mathcal{R}_2$ | 5 2 [POW] $\rightarrow$ 25 |
| [CEIL] | Get $\mathcal{R}, \mathcal{N} \Rightarrow$ Return the value that round up $\mathcal{R}$ from the Nth decimal place | 1.13 2 [CEIL] $\rightarrow$ 1.2 |
| [FLOOR] | Get $\mathcal{R}, \mathcal{N} \Rightarrow$ Return the value that round down $\mathcal{R}$ to the $\mathcal{N}$th decimal place | 34.72 1 [FLOOR] $\rightarrow$ 34.7 |
| [ROUND] | Get $\mathcal{R}, \mathcal{N} \Rightarrow$ Return the value that round $\mathcal{R}$ to the $\mathcal{N}$th decimal place | 22.679 2 [ROUND] $\rightarrow$ 22.68 |
| [ABS] | Get $\mathcal{R} \Rightarrow$ Return Absolute of $\mathcal{R}$ | -13 [ABS] $\rightarrow$ 13 |
| [COMB] | Get $\mathcal{N}_1, \mathcal{N}_2 \Rightarrow$ Return $\mathcal{N}_1$ combination $\mathcal{N}_2$ | 3 2 [COMB] $\rightarrow$ 3 |
| [PERM] | Get $\mathcal{N}_1, \mathcal{N}_2 \Rightarrow$ Return $\mathcal{N}_1$ permutation $\mathcal{N}_2$ | 3 2 [PERM] $\rightarrow$ 6 |
| [GCD] | Get $\mathcal{N}_1, \mathcal{N}_2 \Rightarrow$ Return Greatest Common Divisor of $\mathcal{N}_1, \mathcal{N}_2$ | 6 3 [GCD] $\rightarrow$ 3 |
| [LCM] | Get $\mathcal{N}_1, \mathcal{N}_2 \Rightarrow$ Return Least Common Multiple of $\mathcal{N}_1, \mathcal{N}_2$ | 2 3 [LCM] $\rightarrow$ 6 |
| [LIST_SOL] [LIST_EOL] | Declare start and end of LIST, Used in pairs | [LIST_SOL] 1 2 [LIST_EOL] $\rightarrow$ [ 1 2 ] |
| [LIST_ARANGE] | Get $\mathcal{N}_1, \mathcal{N}_2$ and Step $\mathcal{N}_3 \Rightarrow$ Return arithmetic progression LIST | 0 4 1 [LIST_ARANGE] $\rightarrow$ [ 0 1 2 3 4 ] |
| [LIST_ODD] | Get $\mathcal{N}_1, \mathcal{N}_2 \Rightarrow$ Return odd number list between LIST $\mathcal{N}_1$ and $\mathcal{N}_2$ | 0 4 [LIST_ODD] $\rightarrow$ [ 1 3 ] |
| [LIST_EVEN] | Get $\mathcal{N}_1, \mathcal{N}_2$ numbers $\Rightarrow$ Return even number LIST between $\mathcal{N}_1$ and $\mathcal{N}_2$ | 0 4 [LIST_EVEN] $\rightarrow$ [ 0 2 4 ] |
| [LIST_POP] | Pops the topmost LIST from the stack | |
| [LIST_GET_PERM] | Get $\mathcal{L}$ with $\mathcal{N}$s, $\mathcal{N}_1 \Rightarrow$ Return all possible $\mathcal{N}_1$ digit natural numbers by using $\mathcal{N}$s once | [LIST_SOL] 2 3 [LIST_EOL] 2 [LIST_GET_PERM] $\rightarrow$ [ 23 32 ] |
| [LIST_GET_PRODUCT] | Get $\mathcal{L}$ with $\mathcal{N}$s, $\mathcal{N}_1 \Rightarrow$ Return all possible $\mathcal{N}_1$ digit natural numbers allowing duplicate $\mathcal{N}$s | [LIST_SOL] 2 3 [LIST_EOL] 2 [LIST_GET_PRODUCT] $\rightarrow$ [ 22 23 32 33 ] |
| [GEN_POSSIBLE_LIST] | Get $\mathcal{S}$ with $\mathcal{V}_1, \mathcal{V}_1 \Rightarrow$ Return all possible numbers by substituting 1-9 to $\mathcal{V}_1$ | 3A [GEN_POSSIBLE_LIST] $\rightarrow$ [ 30, ..., 39 ] |
| [LIST_MAX] | Get $\mathcal{L}, \mathcal{N} \Rightarrow$ Sort $\mathcal{L}$ in descending order $\Rightarrow$ Return $\mathcal{N}$th value in sorted $\mathcal{L}$ | [LIST_SOL] 11 10 12 [LIST_EOL] 2 [LIST_MAX] $\rightarrow$ 11 |
| [LIST_MIN] | Get $\mathcal{L}, \mathcal{N} \Rightarrow$ Sort $\mathcal{L}$ in ascending order $\Rightarrow$ Return $\mathcal{N}$th value in sorted $\mathcal{L}$ | [LIST_SOL] 11 10 12 [LIST_EOL] 1 [LIST_MIN] $\rightarrow$ 10 |
| [LIST_SUM] | Get $\mathcal{L} \Rightarrow$ Return sum of the $\mathcal{L}$ | [LIST_SOL] 11 10 12 [LIST_EOL] [LIST_SUM] $\rightarrow$ 33 |
| [LIST_LEN] | Get $\mathcal{L} \Rightarrow$ Return length of the $\mathcal{L}$ | [LIST_SOL] 11 10 12 [LIST_EOL] [LIST_LEN] $\rightarrow$ 3 |
| [LIST_GET] | Get $\mathcal{L}, \mathcal{N}_1 \Rightarrow$ Return $\mathcal{N}_1$th value of $\mathcal{L}$ ($\mathcal{N}_1$ starts from 1) | [LIST_SOL] 11 10 12 [LIST_EOL] 3 [LIST_GET] $\rightarrow$ 12 |
| [LIST_INDEX] | Get $\mathcal{L}, \mathcal{N}_1 \Rightarrow$ Find $\mathcal{N}_1$ in $\mathcal{L}_1$ and Return index of $\mathcal{N}_1$ from $\mathcal{L}$ (index starts from 1) | [LIST_SOL] 11 10 12 [LIST_EOL] 11 [LIST_INDEX] $\rightarrow$ 1 |
| [LIST_FIND_NUM] | Get $\mathcal{L}, \mathcal{N}_1 \Rightarrow$ Return total number of $\mathcal{N}_1$ in $\mathcal{L}$ | [LIST_SOL] 1 2 3 2 1 [LIST_EOL] 1 [LIST_FIND_NUM] $\rightarrow$ 2 |

| [LIST_MORE] | Get $\mathcal{L}, \mathcal{N}_1 \Rightarrow$ Return LIST with $\mathcal{L}$'s elements bigger than $\mathcal{N}_1$ | [LIST_SOL] 11 10 12 [LIST_EOL] 11 [LIST_MORE] $\rightarrow$ 12 |
|---|---|---|
| [LIST_LESS] | Get $\mathcal{L}, \mathcal{N}_1 \Rightarrow$ Return LIST with $\mathcal{L}$'s elements smaller than $\mathcal{N}_1$ | [LIST_SOL] 11 10 12 [LIST_EOL] 11 [LIST_LESS] $\rightarrow$ 10 |
| [LIST_MORE_EQUAL] | Get $\mathcal{L}, \mathcal{N}_1 \Rightarrow$ Return LIST with $\mathcal{L}$'s elements same or bigger than $\mathcal{N}_1$ | [LIST_SOL] 11 10 12 [LIST_EOL] 11 [LIST_MORE_EQUAL] $\rightarrow$ [11 12] |
| [LIST_LESS_EQUAL] | Get $\mathcal{L}, \mathcal{N}_1 \Rightarrow$ Return LIST with $\mathcal{L}$'s elements same or smaller than $\mathcal{N}_1$ | [LIST_SOL] 11 10 12 [LIST_EOL] 11 [LIST_LESS_EQUAL] $\rightarrow$ [11 10] |
| [SET_UNION] | Get $\mathcal{L}_1, \mathcal{L}_2 \Rightarrow$ Return LIST which is union of $\mathcal{L}_1, \mathcal{L}_2$ | [LIST_SOL] 1 2 3 [LIST_EOL] [LIST_SOL] 2 3 4 [LIST_EOL] [SET_UNION] $\rightarrow$ [ 1 2 3 4 ] |
| [SET_DIFFERENCE] | Get $\mathcal{L}_1, \mathcal{L}_2 \Rightarrow$ Return $\mathcal{L}_1$ differnt from $\mathcal{L}_2$ | [LIST_SOL] 1 2 3 [LIST_EOL] [LIST_SOL] 2 3 4 [LIST_EOL] [SET_DIFFERENCE] $\rightarrow$ 1 |
| [SET_INTERSECT] | Get $\mathcal{L}_1, \mathcal{L}_2 \Rightarrow$ Return LIST which is intersect of $\mathcal{L}_1, \mathcal{L}_2$ | [LIST_SOL] 1 2 3 [LIST_EOL] [LIST_SOL] 2 3 4 [LIST_EOL] [SET_INTERSECT] $\rightarrow$ [ 2 3 ] |
| [LIST_DIVISIBLE] | Get $\mathcal{L}, \mathcal{N}_1 \Rightarrow$ Return $\mathcal{L}$'s element which is divisible by $\mathcal{N}_1$ in LIST | [LIST_SOL] 4 10 12 [LIST_EOL] 4 [LIST_DIVISIBLE] $\rightarrow$ [ 4 12 ] |
| [LIST_DIVIDE_AND _REMAIN] | Get $\mathcal{L}, \mathcal{N}_1, \mathcal{N}_2 \Rightarrow$ Return LIST of $\mathcal{L}$'s element if the remainder of dividing $\mathcal{L}$'s element by $\mathcal{N}_1$ is $\mathcal{N}_2$. | [LIST_SOL] 4 7 12 [LIST_EOL] 3 1 [LIST_DIVIDE_AND_REMAIN] $\rightarrow$ [ 4 7 ] |
| [LIST_GET_DIVISOR] | Get $\mathcal{N} \Rightarrow$ Return LIST of divisors of $\mathcal{N}$ | 6 [LIST_GET_DIVISOR] $\rightarrow$ [ 1 2 3 6 ] |
| [LIST_COND_MAX _MIN] | Get $\mathcal{L}_1, \mathcal{L}_2 \Rightarrow$ Return LIST in which $\mathcal{L}_1$ is sorted in descending order according to the condition of $\mathcal{L}_2$ | [LIST_SOL] (A) (B) (C) (D) [LIST_EOL] [LIST_SOL] (B) (D) > (B) (A) > (C) (B) > [LIST_EOL] [LIST_COND_MAX_MIN] $\rightarrow$ (C) (B) (D) (A) |
| [LIST2NUM] | Get $\mathcal{L} \Rightarrow$ Return $\mathcal{N}$ with each number of digits in order of $\mathcal{L}$'s element | [LIST_SOL] 1 2 3 [LIST_EOL] [LIST2NUM] $\rightarrow$ 123 |
| [NUM2LIST] | Get N $\Rightarrow$ Return LIST of the digits of each digit of $\mathcal{N}$ | 123 [NUM2LIST] $\rightarrow$ [ 1 2 3 ] |
| [LIST_NUM2SUM] | Get $\mathcal{L} \Rightarrow$ Return LIST of each $\mathcal{L}$'s element's sum of the digits | [LIST_SOL] 10 23 77 [LIST_EOL] [LIST_NUM2SUM] $\rightarrow$ [ 1 5 14 ] |
| [LIST_SEARCH_FIXED _DIGIT] | Get $\mathcal{L}, \mathcal{N}_1, \mathcal{N}_2 \Rightarrow$ Return LIST, which has only elements of $\mathcal{L}$ with Return $\mathcal{N}_1$th digit are Return $\mathcal{N}_2$. | [LIST_SOL] 112 14 32 [LIST_EOL] 2 1 [LIST_SEARCH_FIXED_DIGIT] $\rightarrow$ [ 112 14 ] |
| [DIGIT_UNK_SOLVER] | Get $\mathcal{E}, \mathcal{V} \Rightarrow$ Return $\mathcal{N}$, which is the value of $\mathcal{V}$, satisfies $\mathcal{E}$. If there are multiple values that can fit in $\mathcal{V}$, return LIST. $\mathcal{V}$ is unit digit. | A2+4B=69 A [DIGIT_UNK_SOLVER]$\rightarrow$ 2 |
| [LIST_FIND_UNK] | Get $\mathcal{L}, \mathcal{S}, \mathcal{V} \Rightarrow$ Return $\mathcal{N}$, which is the value of $\mathcal{V}$. If there are multiple values that can fit in $\mathcal{V}$, return LIST. It should be used as a pair with [GEN_POSSIBLE_LIST] | 3A [GEN_POSSIBLE_LIST] 3A A [LIST_FIND_UNK]$\rightarrow$ [ 0, ..., 9 ] |
| [NUM_UNK_SOLVER] | Get $\mathcal{E}, \mathcal{S} \Rightarrow$ Return $\mathcal{N}$, which is the value of S, satisfies $\mathcal{E}$. If there are multiple values that can fit in $\mathcal{S}$, return LIST. $\mathcal{S}$ is not a unit digit but a string. | 3/7=6/A=B/21 A [NUM_UNK_SOLVER]$\rightarrow$ 14 |

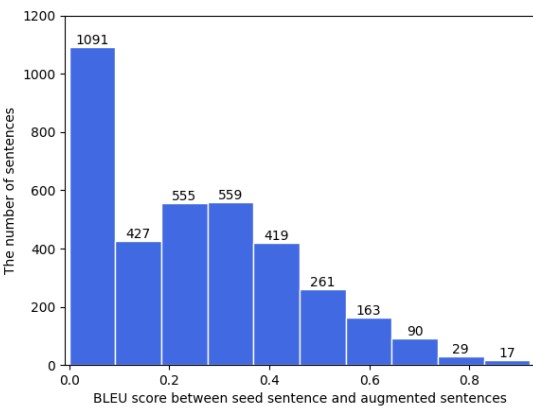

Figure 3: The histogram of BLEU score between seed sentences and augmented sentences.

## C   Statistics for BLEU score

The histogram for BLEU score between seed sentences and augmented sentences is in Figure 3. The statistics for BLEU score between seeds and augmented problems are here: 0.0 for min, 0.24 for mean, 0.23 for median, and 0.92 for max.

## D   Characteristics of other benchmkark datasets

A model can only be considered capable of reasoning if it is capable of elementary school-level logical reasoning. The mathematical reasoning abilities are diverse, as shown Section 2; however, the current MWP task is focused on arithmetical reasoning using four fundamental arithmetic operations (Koncel-Kedziorski et al., 2015; Roy and Roth, 2017; Miao et al., 2020; Cobbe et al., 2021).

Data such as MATH (Hendrycks et al., 2021) or MathQA (Amini et al., 2019) have other reasoning problems. But MATH is too domain-specific. As mentioned in Liang et al. (2016), we think the MWP task is a good problem in evaluating the ability to understand and reason because natural language narratives of grade school-level math problems have less combined syntax and contain a small amount of domain knowledge. Our objective was to develop an elementary school-level (grades 1-6) math dataset, whereas MATH caters to high-school-level (grades 8-12) mathematics. It encompasses intricate mathematical problems from AMC10, AMC12, and AIME that necessitate comprehension of advanced mathematical concepts such as trigonometric functions or imaginary numbers. To ensure a fair comparison, we compared it with grade school-level datasets such as GSM8K, SVAMP, and ASDiv.

Also, MathQA has too low a lexical diversity (Miao et al., 2020). It has been modified several times by Austin et al. (2021) and Jie et al. (2022) but still not enough. We inspect 20 problems that GPT-4 fails on in modified MathQA dataset (Austin et al., 2021) and find out that 30% of problems have incorrect annotation, 10% of problems have vague problems, and 5% of problems are unsolvable. Miao et al. (2020) got similar results, saying that 67% of problems have incorrect formulas, 23% of problems have problematic descriptions, and 10% of problems have valueless answers, randomly inspecting 30 inconsistent samples. So, we choose only GSM8K (Cobbe et al., 2021), SVAMP (Patel et al., 2021), and ASDiv (Miao et al., 2020) only.

## E   MWP solving framework for fine-tuning methods

We adopt a general framework to evaluate MWP models. It consists of three modules: *template-based conversion*, *MWP model*, and *equation-to-code conversion*.

First, we convert some tokens in the narrative to pre-defined symbol tokens. Using a rule-based template method, we only transform pre-defined token values into specific symbols. For instance, the numerical tokens in the narrative are converted to $[N_1], [N_2], \ldots, [N_x]$. Besides, some pre-defined tokens are converted to the elements in the list.

Second, the MWP models are trained to translate a narrative with abstract symbols into a postfix equation. Although many existing studies have been proposed, (Xie and Sun, 2019; Zhang et al., 2020; Jie et al., 2022) some models are limited to handling various operators. In other words, these models are designed to solve the binary operator data like $+, -, \times, \div$. DMath has many operators that are not binary. So, in this paper, we employ MWP models using pre-trained language models, *i.e.*, RoBERTa (Liu et al., 2019) and GPT-2 (Radford et al., 2019).

Lastly, we convert the postfix expression to Python code. Because the operators are pre-defined, we can generate the executable Python code from the postfix expression. When the postfix expression is imprecise, it may fail to convert the postfix expression to the Python code. We simply return a NULL value as the answer for the exceptional case. When we execute the Python code, we utilize the values for the token mapping as the argument of the Python code.

## F  Prompt examples

In this section, we describe prompting methods used in DMath. The zero-shot (Brown et al., 2020) prompting is written to give a question and generate an answer immediately, and the zero-shot CoT (Kojima et al., 2022) prompting is written to generate the answer after giving the question and 'Let's think step by step'. Prompt examples for zero-shot and zero-shot CoT per language can be found in Table 9.

| | | |
|---|---|---|
| **Zero-shot** | **EN** | Q: {problem}
A: The answer is |
| | **KO** | 질문: {problem}
답변: 답은 |
| **Zero-shot CoT** | **EN** | [1 step]
Q: {problem}
A: Let's think step by step.

[2 step]
Q: {problem}
A: Let's think step by step. {1 step output} The answer is |
| | **KO** | [1 step]
질문: {problem}
답변: 문제에 대해 단계별로 풀어보자.

[2 step]
질문: {problem}
답변: 문제에 대해 단계별로 풀어보자. {1 step output} 답은 |

Table 9: Prompt examples per language for zero-shot (Brown et al., 2020) and zero-shot CoT (Kojima et al., 2022).

In section 5, we use *one category random* method as examples of few-shot CoT (Wei et al., 2022) in DMath. The following table shows the few-shot examples of *one category random*. When using the babbage, curie, and davinci models, the prompting before the target question was truncated if few-shot CoT prompting exceeds 2049 tokens.

**Q**: You sold 40 toys on Day 1. On Day 2, you sold 4 times the number of toys sold on Day 1. Find out the number of toys you sold more on Day 2 compared to Day 1.
**A**: On the Day 1, you sold 40 toys. On the Day 2, you sold 4 times more toys than on the Day 1, so you sold 4 * 40 = 160 toys. Therefore, the Day 2 sold 160 - 40 = 120 more toys than the Day 1. Therefore the answer is 120.

**Q**: There is a banner 20 meters (m) long. I'm thinking of fixating this banner with nails. Nails were driven 1 meter (m) apart from both ends of the banner, and the remaining nails were nailed in at regular intervals. A total of 7 nails were used. Find the distance between one nail and another.
**A**: The interval between the first and last nails is 20 - 1 * 2 = 18. Since there are a total of seven nails, there are 7 - 1 = 6 gaps between the nails. Therefore, the interval between nails is 18 / 6 = 3. Therefore the answer is 3.

**Q**: Yeongchan's group consists of 7 people. One person can fold two paper cranes in one minute. How many minutes will it take Yeongchan's group to fold 182 paper cranes?
**A**: In Yeongchan's group, one person can fold two paper cranes per minute, and in Yeongchan's group, there are seven people, so Yeongchan's group can fold 2 * 7 = 14 paper cranes per minute. If Youngchan's group folds 182 paper cranes, it takes 182 / 14 = 13 minutes. Therefore the answer is 13.

**Q**: Namjoon rode his bike and traveled 96.967 kilometers (km) in 1.3 hours. If Namjoon went at a steady speed, how many kilometers (km) did he travel in 1 hour?
**A**: The distance Namjoon went for an hour is 96.967 / 1.3 = 74.59 kilometers (km). Therefore the answer is 74.59.

**Q**: There are 24 female and male students in total. Among them, 50 sheets of colored paper were equally distributed to female students only, and 2 sheets were left. How many pieces of colored paper will the girls have if there are twice as many boys as girls?
**A**: Female students have 50 - 2 = 48 colored papers. When the number of girls is x, the number of boys is 2x. Since there are 24 female and male students, it is x + 2x = 3x = 24. Therefore, the number of female students is 8. One female student has 48 / 8 = 6. Therefore the answer is 6.

**Q**: There is a train that runs at 95 kilometers (km) per hour and a car that runs at 65 kilometers (km) per hour. If the train and the car travel for 8 hours at a constant speed, by how many kilometers (km) does the train travel more than the car?
**A**: The train traveled 95 * 8 = 760 kilometers (km) in 8 hours, and the car traveled 65 * 8 = 520 kilometers (km) in 8 hours. Therefore, the train went 760 - 520 = 240 kilometers (km) longer than the car. Therefore the answer is 240.

**Q**: There are a total 12 of 10-won, 50-won, and 100-won coins. I checked the total amount and it was 500 won. If the number of 10-won and 50-won coins is the same, how many 100-won coins are there?
**A**: Let's say the number of 10-won coins and 50-won coins is x and the number of 100-won coins is y. If so, there are 12 coins worth 10, 50, and 100-won, so x + x + y = 12. Since the total amount is 500-won, it is 10 * x + 50 * x + 100 * y = 500. Solving this equation, x is 5 and y is 2. Therefore the answer is 2.

**Q**: There is a scale that can weigh more than 2 kilograms (kg). If you have lifted 3 weights of 200 grams (g) so far, find how many more weights you need to lift to measure the weight.
**A**: I've put up three 200 grams (g) weights, so there's an 200 * 3 = 600 grams (g) on the scale now. 2 * 1000 - 600 = 1400 grams (g) is needed to weigh 2 kilograms (kg) more. Therefore, 1400 / 200 = 7 additional 200 grams (g) scales are needed. Therefore the answer is 7.

**Q**: {problem}
**A**:

**Q**: There are 24 balloons. Of these, there are 6 more blue balloons than red balloons, red balloons are 1/4 of the total, and the rest are all yellow balloons. What color balloons will be the most?

**A**: Since the red balloons are 1/4 of the total, 24 * ( 1 / 4 ) = 6 balloons. Blue balloons are 6 + 6 = 12 because there are 6 more blue balloons than red ones. The rest are yellow balloons, so 24 - 6 - 12 = 6. Therefore, blue balloons are the most common. Therefore the answer is blue.

**Q**: How many of 0.8, 1/2, and 0.9 are greater than 0.4?

**A**: Of 0.8, 1/2, and 0.9, three are greater than 0.4. Therefore the answer is 3.

**Q**: Yoongi has 4 apples and Jungkook has 6 divided by 3 apples. Who has the greater number of apples?

**A**: Jungkook has 6 / 3 = 2 apples because he has 6 divided by 3 apples. Yoongi has four apples, so he has more apples than Jungkook. Therefore the answer is Yoongi.

**Q**: Choose the 2nd smallest number among 5, 8, 4, 3, and 2.

**A**: The second smallest number of 5, 8, 4, 3, and 2 is 3. Therefore the answer is 3.

**Q**: Eight students stand in a line to borrow books. How many students are standing between the student standing 1st from the front and the student standing 4th from the back?

**A**: When a total of eight students are standing in a line, there are 8 - 1 - 4 = 3 students between the first student in the front and the fourth student in the back. Therefore the answer is 3.

**Q**: Taehyung, Minju, Sangmin, Yoonjung, and Yoojeong took turns crossing the finish line. Who went in first?

**A**: Taehyung is first place. Therefore the answer is Taehyung.

**Q**: Find the number of numbers less than or equal to 0.4 among 0.8, 1/2, and 0.3.

**A**: The number less than or equal to 0.4 of 0.8, 1/2, and 0.3 is 0.3. Therefore the answer is 1.

**Q**: What is the smallest number among the numbers greater than 1.1 among the numbers 1.4, 9/10, 1.2, 0.5, and 13/10?

**A**: Among 1.4, 9/10, 1.2, 0.5, and 13/10, the number greater than 1.1 is 1.4, 1.2, and 13/10. The smallest number of these is 1.2. Therefore the answer is 1.2.

**Q**: {problem}

**A**:

**Q**: There are three natural numbers A, B, and C. When A is divided by 7, the quotient is B and the remainder is C. The quotient and remainder are equal in this equation. Find the largest number among A.

**A**: When A is divided by 7, the remaining C can be 0, 1, 2, 3, 4, 5, and 6. In order to make A the largest, C must be 6 and B must be 6. Thus, the largest A is 6 * 7 + 6 = 48. Therefore the answer is 48.

**Q**: There is a single digit number A. When 7A7 is rounded up to the tens place is 730. What is the right value for A?

**A**: The fact that 7A7 is raised from the decimal place to 730 means that A is 2. Therefore the answer is 2.

**Q**: Jisoo had 6,000 won and she bought a ballpoint pen for 3,200 won, an eraser for 1,000 won, and candy for 500 won. Assuming that all remaining money is saved, find the amount saved.

**A**: The amount of money Jisoo has left after buying the product is 6000 - 3200 - 1000 - 500 = 1300 won. Therefore the answer is 1300.

**Q**: The length of Siwon's ribbon is 8.8 meters longer than Seojun's. If Seojun cuts his own ribbon 4.3 meters (m) and gives it to Siwon, find out what the difference between the lengths of the two people's ribbons.

**A**: If Siwon's ribbon length is x, Seojun's ribbon length is x - 8.8. After Seojun gives the ribbon to Siwon, the length of Seojun's ribbon is x - 8.8 - 4.3, and the length of Siwon's ribbon is x + 4.3. The difference between the two is ( x + 4.3 ) - ( x - 8.8 - 4.3 ) = 17.40 Therefore the answer is 17.40.

**Q**: 37 is 69 smaller than a number. What is the number that is 55 greater than the number?

**A**: A number that is 69 smaller than the number is 37, so the number is 69 + 37 = 106. Therefore, a number greater than the number by 55 is 106 + 55 = 161. Therefore the answer is 161.

**Q**: 24 is a result of subtracting 63 from a particular number by mistake when the original calculation was to subtract 36 from that particular number. Find the result of the original calculation.

**A**: The number is 24 + 63 = 87. Correctly calculated, the value obtained by subtracting 36 from the number is 87 - 36 = 51. Therefore the answer is 51.

**Q**: It was 40 that should be subtracted from, but mistakenly subtracting 21 from a number yields 52. How much do you get when you calculate it correctly?

**A**: The number is 52 + 21 = 73. Correctly calculated, subtracting 40 from the number is 73 - 40 = 33. Therefore the answer is 33.

**Q**: You need to subtract 46 from a certain number, but mistakenly subtracted 64, and got 122. Find the result of the correct calculation.

**A**: The number is 122 + 64 = 186. To calculate correctly, subtract 46 from the number is 186 - 46 = 140. Therefore the answer is 140.

**Q**: {problem}
**A**:

**Q**: There is a parallelogram that has a base of 3.6 and a height of 2.5 times the base. Find the area of this parallelogram.

**A**: The area of the parallelogram is the product of the base and the height. Since the base of this parallelogram is 3.6 and the height is 3.6 * 2.5 = 9, the area is 3.6 * 9 = 32.4. Therefore the answer is 32.4.

**Q**: 12 friends, including Yoonseok, are standing as a dodecagon. Every student shook hands once with every friend except for the friend on either side of them. How many handshakes did Yoonseok shake?

**A**: Yoon Seok shook hands 12 - 2 - 1 = 9 times because he shook hands with friends except for two on both sides of him and one on his own. Therefore the answer is 9.

**Q**: There are pretty blankets whose shape is the same as a regular tetradecagon. When the length of all corners is 154 centimeters (cm). Find the length of one edge of the blanket.

**A**: Since there are a total of 14 edges of a square, the length of one edge is 154 / 14 = 11. Therefore the answer is 11.

**Q**: How many square centimeters ($cm^2$) is the sum of the widths of two squares, every 11 centimeters (cm) and 5 centimeters (cm) long?

**A**: The area of a square with a side length of 11 centimeters (cm) is 11 * 11 = 121 square centimeters ($cm^2$), and the area of a square with a side length of 5 centimeters (cm) is 5 * 5 = 25 square centimeters ($cm^2$). The sum of the widths of the two squares is 121 + 25 = 146 square centimeters ($cm^2$). Therefore the answer is 146.

**Q**: The perimeter of the square is 17.8 centimeters (cm). Find the length of one side of this square.

**A**: The circumference of a square is four times the length of a side. Therefore, the length of one side of a square with a circumference of 17.8 centimeters (cm) is 17.8 / 4 = 4.45 centimeters (cm). Therefore the answer is 4.45.

**Q**: What is the number of edges of a cube?

**A**: The number of edges of a cube is 6 * 2 = 12. Therefore the answer is 12.

**Q**: After arranging several cans of soda in a square shape, there were 6 cans left. So, I tried to increase the width and length by one row, but 11 cans were inadequate. If you arrange these cans in 7 rows, how many cans are there in one row?

**A**: Let's say that the number of cans of soda currently on one side of the square is x. If you want to increase the width and length by one row, you need 2 * x + 1 cans of soda. Currently, there are 6 cans left, and 11 cans are insufficient when trying to increase it, so it can be considered 2 * x + 1 = 6 + 11. So x is 8. The number of beverage cans currently in possession is x * x + 6 = 8 * 8 + 6 = 70. If these 70 cans are set in seven rows, there are 70 / 7 = 10 cans in the first row. Therefore the answer is 10.

**Q**: There is a rhombus with an area of 64/5 square centimeters ($cm^2$). If one diagonal is 64/9 centimeters (cm), how long is the other diagonal in centimeters (cm)?

**A**: The area of the rhombus is equal to the product of the two diagonals in the rhombus multiplied by one-half. Thus, when one diagonal is 64/9 centimeters (cm), the length of the other diagonal is (64 / 5) * 2 / (64 / 9) = 3.6 centimeters (cm). Therefore, the answer is 3.6.

**Q**: {problem}
**A**:

**Q**: There are two types of Korean language workbooks and four types of math workbooks. When choosing one book among them, in how many cases can you choose Korean workbook or math workbook?

**A**: When there are two types of Korean workbooks and four types of math workbooks, the number of cases in which one Korean workbooks or one math workbooks is selected is 2 + 4 = 6. Therefore the answer is 6.

**Q**: You want to create a three-digit number by drawing three different numbers from 0, 1, 3, and 5. Find the sum of the largest and smallest possible numbers.

**A**: The largest number that can be made is 531, and the smallest number is 103. The sum of the two is 531 + 103 = 634. Therefore the answer is 634.

**Q**: Nine digits from 1 to 9 are used once to form a nine-digit number. If this number is a multiple of 55, find the largest 9-digit number.

**A**: The largest number of nine digits, a multiple of 55, is 987642315. Therefore the answer is 987642315.

**Q**: How many two-digit numbers can be formed by selecting two different numbers from 1, 3, and 4?

**A**: The number of two digits that can be made by drawing two different numbers from 1, 3, and 4 is 3P2 = 6. Therefore the answer is 6.

**Q**: There are 5 marbles: red, blue, yellow, purple, and black. You want to choose two of these and give one to your younger brother and the other to your older brother as a gift. Find the number of cases in this situation.

**A**: The number of cases in which two of the five different colored beads are selected is 5P2 = 20. Therefore the answer is 20.

**Q**: You have 4 blue pencils and 1 red pencil. How many different ways can you place 5 pencils in a row?

**A**: When there are four blue pencils and one red pencil, there are a total of five ways to put these five pencils in one line. Therefore the answer is 5.

**Q**: How many multiples of 4 are there among the four-digit numbers made using the number cards 1, 3, 5, and 6 once?

**A**: The four digits that can be made using the number cards 1, 3, 5, and 6 once are 1356, 1365, 1536, 1563, 3156, 3165, 3516, 3516, 3561, 3615, 3651, 5136, 5163, 5163, 5361, 5613, 6135, 6153, 6351, 6513, 6531, 6531. Among them, the multiples of 4 are 1356, 1536, 3156, 3516, 5136, and 5316. Therefore the answer is 6.

**Q**: Find the smallest number that can be made by using all the number cards 0, 2, 4, 6, 8 and 9 only once.

**A**: The smallest number that can be made using the number cards of 0, 2, 4, 6, 8, and 9 once is 204689. Therefore the answer is 204689.

**Q**: {problem}
**A**:

# G  Failed examples in GPT-4

## ARI

**Question:** How many four-digit numbers are divisible by the two numbers 3 and 5?

**Reasoning:** A number that is divisible by both 3 and 5 is divisible by their least common multiple (LCM). The LCM of 3 and 5 is 15. To find the number of four-digit numbers divisible by 15, we need to find the range of these numbers. The smallest four-digit number is 1000, and the largest is 9999. The smallest four-digit number divisible by 15 is 1005 (1000 + 5), and the largest is 9990 (9999 - 9). Now, we need to find how many numbers are in this range that are divisible by 15. To do this, we can divide the difference between the largest and smallest numbers by 15 and add 1. (9990 - 1005) / 15 + 1 = 5985 / 15 + 1 = 399 + 1 = 400. So, there are 400 four-digit numbers that are divisible by both 3 and 5.

**Correct Answer:** 600

## COM

**Question:** Nine people stand in a line in order from shortest to tallest. Hoseok stands in the front. If they line up again in order from the tallest to the shortest, what number will Hoseok stand from the back?

**Reasoning:** Since Hoseok is the shortest and stands in the front when they line up from shortest to tallest, he will be the last person when they line up from tallest to shortest. So, Hoseok will stand 9th from the back. Therefore the answer is 9.

**Correct Answer:** 1

## COR

**Question:** When you round 34AC to the nearest hundred, you get 3400. Find the sum of the possible As.

**Question:** When rounding 34AC to the nearest hundred, the possible values for A are 3, 4, 5, and 6. This is because if A is 3 or 4, the number will round down to 3400, and if A is 5 or 6, the number will round up to 3400. The sum of the possible As is 3 + 4 + 5 + 6 = 18. Therefore, the answer is 18.

**Correct Answer:** 10

## GEO

**Question:** How many sides are there in a polygon which 12 diagonals can be drawn from one vertex?

**Reasoning:** If 12 diagonals can be drawn from one vertex, then there are 12 other vertices that are not adjacent to the given vertex. Since a polygon has as many sides as it has vertices, there must be 12 + 1 (the given vertex) = 13 sides in the polygon. Therefore, the polygon has 13 sides.

**Correct Answer:** 15

## POS

**Question:** You want to create a three-digit number by drawing three cards out of number cards with 1, 7, 0, and 3 written on them and using them only once. Find the value of multiplying the second largest number and the second smallest number that can be made.

**Reasoning**: The second largest number that can be made is 371, and the second smallest number is 130. The value of multiplying the two is 371 × 130 = 48,230. Therefore the answer is 48,230.

**Correct Answer:** 78110