# OpenReview forum: "It Ain't Over: A Multi-aspect Diverse Math Word Problem Dataset"
_EMNLP/2023/Conference — EMNLP 2023 Main_

### Official Review · Reviewer_uqoi · 2023-08-03

**Soundness:** 4

**Excitement:**

4: Strong: This paper deepens the understanding of some phenomenon or lowers the barriers to an existing research direction.

**Paper Topic And Main Contributions:**

This paper introduces a new MWP dataset, DMath, which is diverse in lexicon usages and problem types. It is also bilingual (English and Korean languages) and annotated two kinds of solutions (an expression tree and Python code). Experimental results illustrate that the GPT-4 LLM achieves 70% accuracy on DMath; the performance gap is worth exploring for the future research on MWP solving.

**Questions For The Authors:**

Q1: Table 2: Is P(3,2) a more precise and comprehensive expression for human annotating (for Human solution, Equation and Python code) and understanding?

Q2: Lines #163-#164: Why didn’t DMath provide the validation set?

Q3: Table 6: The performance of the few-shot CoT is lower than the zero-shot Cot for the CGP-4 model on GSM-8K. Why? How to obtain the few-shot samples of SVAMP and ASDIV?

Q4: Lines #253-#265: How to verify and confirm steps (1)~(3)? The validation process is essential for an MWP dataset. It is suggested to provide the validation details. For step (4), it is unclear how to “make the expression tree as efficient as possible”, and “be acceptable to most people”? It is also suggested to provide detailed information.

Q5: Lines #277-#279: How do the remaining 71% ensure grammatical correctness?

**Reasons To Accept:**

This paper introduces a new dataset. It not only focuses on the diversity of the language pattern and problem type but also provides bilingual MWPs and multiple solution reasoning paths. Observations, insights, and discussions of relevant experimental results are detailly and will be helpful for future research on MWP solving. Furthermore, most previous studies of non-LLM models are limited to binary operators, so the proposed dataset (including many non-binary operators) will help future research on MWP solving.

**Reasons To Reject:**

(1) Lack of detailed description of the validation procedures of DMath (refer to Q3). Therefore, there are potential risks to the correctness of the annotated data.

(2) The intermediate solutions (including Human solutions and Equations) are annotated by pre-defined intermediate form instead of mathematical languages. Such annotations may be hard to interpret and comprehend directly by the human.

**Reproducibility:**

3: Could reproduce the results with some difficulty. The settings of parameters are underspecified or subjectively determined; the training/evaluation data are not widely available.

**Reviewer Confidence:**

4: Quite sure. I tried to check the important points carefully. It's unlikely, though conceivable, that I missed something that should affect my ratings.

---

> ### Author Rebuttal · Authors · 2023-08-29
>
> Saying that our paper will help with future research is appreciated. Also, thank you for pointing out many things that will help our paper. The responses are below.
>
> ## Reasons to Reject
> ### 1. Lack of detailed description of the validation procedures of DMath.
> * **We agree that a validation procedure for MWP datasets is very important.** For this purpose, we recruited 13 graduate students majoring in computer science who participated in the previous process (*collection, augmentation*, and *annotation*). They are familiar with the annotation format of the MWP dataset and can solve elementary-level mathematical problems well.
>
> * **We tried to elaborate the validation process and performed four processes sequentially:**
>
> > **1. Question-answer validation,**
> > **2. Question-expression tree validation,**
> > **3. Expression tree validation, and**
> > **4. Conciseness validation of expression tree for multi-expression-tree matching problems.**
>
> * If any type of error \(*i.e. errors in the question, in the expression tree*, or *in the answer*\) was detected during \(1\)-\(2\)-\(3\), we returned to \(1\) for revalidation. After that, we performed (4\). **These procedures were done to reduce potential risks to the correctness of the annotated data.**
> * Here's how the human workers were allocated for each step:
>   * Across step (1) and step (2), we ensured that the questions assigned to a human worker do not overlap so that they do not receive the same questions in each step, which can act as a cross-check.
>   * Step (3) was verified using code.
>   * For step (4), 2-3 human workers were assigned per question.
>
> * Now we elaborate the details of steps (1)~(4) on Lines \#253-\#265.
> #### **Step (1) - \[Question-answer validation\]**
> * After the *annotation* process, we gave human workers only natural language questions and asked them to solve and answer them manually. There are two cases where their answers differ from the annotated answers.
> * The first is when the natural language problem is unsolvable, and the second is when the answer is incorrect. When a worker encounters these errors while solving a problem, the worker corrects the error manually.
> * In the first case, the worker should correct the natural language question, and in the second case, the worker should correct the annotated answer. In addition, the corrected questions are cross-checked by two or more workers to ensure that there are no errors.
>
> #### **Step (2) - \[Question-expression tree validation\]**
> * Natural language questions that are solvable and correctly annotated with correct answers are given to human workers along with an expression tree. The human workers check whether the given natural language question and the expression tree are correctly paired. If not, they modify the expression tree to match the natural language problem.
>
> #### **Step (3) - \[Expression tree validation\]**
> * After the natural language question and the expression tree have been properly matched in the previous step, we verify that the expression tree produces the correct answer. In DMath, the operators used in expression trees can be converted to Python code. Therefore, after converting each expression tree to code, we check whether the result obtained when executing it is the correct answer. If the two values are different, return to step (1).
>
>
> * With these three steps, we wanted to reduce the errors in our dataset. After validating this, we wanted to make the expression tree more efficient.
>
> #### **Step (4) - \[Conciseness validation of expression tree for multi-expression-tree matching problems\]**
> * There can be various potential solutions to a mathematical problem, and we maintain that the selection of an expression tree is crucial to the model’s training.
> * For example, the following math word problem has multiple expression trees.
> > “There are six people. Everyone shook hands with each other once. Find the total number of handshakes.”
> * We can make the expression tree as one of
> ```
> 1) {( 6 - 1 ) * 6} / 2
> 2) 6 C 2
> ```
> * Here, both equations lead to the correct answer and they are  reasonable.
> * We wondered which of these multi-expression-trees would be better to choose. We decided to choose a problem with as short an equation length as possible. We considered this direction, reducing the equation length, because the most concise solution process captures the essence of the problem.
> * However, even if a formula is not the shortest, we choose it if most workers agree. We made sure that discussions about the most appropriate formulas involved at least three students and an odd number of students to prevent the majority of each question from being 1:1.
> * We also felt that "efficient" was a word that could encapsulate the process.
>
> - We believe that the validation process is important to reduce errors in the dataset, but due to the paper limitation, we have to compress the content and as a result not all of the details are revealed. We apologize for this and will add these details to the paper.
>
> ### 2. Necessity of pre-defined intermediate form
> * **We had to define predefined intermediate forms to solve problems that cannot be solved using only four arithmetic operators.** Typical problems that cannot be solved by using only arithmetic operators are following:
> > 1. Comparing values between each other
> > 2. Number of Cases
> > 3. Problems involving rounding
> > 4. Problems involving unknowns
> * We have defined operators to solve these types of problems, and **we agree that they can be difficult for others to interpret/understand if not well designed.** Therefore, we set some rules for defining these operators.
>     1. Make the names of operators match the names of actual math symbols as closely as possible.
>     *e.g.* `[ADD]`, `[FLOOR]`, `[GCD]`, `[LIST_MAX]`
>     2. When performing the same action, match the name of the operators.
>     *e.g.* \(a\) `[LIST_GET_PERM]` \(b\) `[LIST_GET_PRODUCT]` \(c\) `[LIST_GET_DIVISOR]`
>     \(a\), \(b\), and \(c\) all take a LIST as input and return a LIST as output through some operation \(PERM, PRODUCT, DIVISOR\), so `‘LIST_GET_’` overlaps.
>     3. Make the name of the operators as concise as possible.
>     *e.g.* `[COMB]`, `[LCM]`, `[LIST_LEN]`, `[LIST2NUM]`
>     4. If a single operator performs multiple operations, give it a name that summarizes the operations.
>     *e.g.*  `[DIGIT_UNK_SOLVER] ⇒ A2+4B=69 A [DIGIT_UNK_SOLVER] → 2`
>     This operator takes an expression, string corresponding to the digits in the expression, and finds the unknown numeric value \(UNK\) of the string.
> * To reduce confusion, we made sure that the functions of complex operators, especially those involving LISTs \(e.g., `[LIST_COND_MAX_MIN]`, given a list `L1` and a list `L2` with a condition containing the case of the elements of `L1`, return a new list of `L1` sorted in descending order according to the condition\), were well explained to everyone involved in data annotation and validation. **To help with the interpretation, we created a table of “operator name - description including the form of input and output - example”.** This table can be found in Appendix B.
>
> ## Questions For the Authors
> ### 1. Effective annotation form for Table 2
> * Of course, **`P(3,2)` is a correct solution, and it is much simpler and clearer than `len([apple, peach, pear]) permutation 2`.** However, our objective was to create a solution that emphasizes the logical progression step-by-step.
> * Let’s consider the example problem in Table 2 again. The problem was as follows:
>
> > “You want to pick 2 fruits out of an apple, a peach, a pear and place them in different fruit baskets. What is the Number of Cases?”
>
> - In this case, **the number 3 is not directly present in the problem**, so we have included the process of obtaining 3 in the solution and represented it as `len([apple, peach, pear]) permutation 2`, instead of using permutation.
>
>
> ### 2. Validation set of DMath
> * We did not separate the validation set following GSM8K, which is currently the most widely used in MWP tasks.
>
> ### 3. Lower performance of the few-shot CoT than the zero-shot CoT for GPT-4 model on GSM8K
>
> * Table 6 shows the experimental results of ChatGPT and GPT-4’s zero-shot CoT and few-shot CoT for GSM8K, SVAMP, ASDiv, and DMath. Here, the performance of zero-shot CoT is higher than that of few-shot CoT only when using the GPT-4 model for GSM8K, so it is quite natural to question it.
> * \[1\] states that they used GSM8K data to train GPT-4. **We believe that using GSM8K data improved the zero-shot CoT performance of the GSM8K in GPT-4, and the few-shot examples used previously acted as noise in the trained model.** There are two reasons for this.
>   * The first is that for ChatGPT without GSM8K as training data, the performance of the zero-shot CoT is lower than that of the few-shot CoT.
>   * The second is that the performance varies depending on the few-shot example. This is also shown in \[2\], where the performance of chain-of-thought prompting on GSM8K using LaMDA 137B is varies for different few-shot examples.
> * We would like to clarify at this point **that our experimental results are not erroneous**: the results of the GSM8K GPT-4’s few-shot CoT in Table 6 are similar to the 92% (with 5-shot, chain-of-thought) reported in the GPT-4’s technical report \[1\], so we believe that our experiments are correct. (The results of the zero-shot CoT are not reported here.)
>
>
>
> ### 4. Few-shot samples of SVAMP and ASDiv
> * **We use the same few-shot example as GSM8K for ASDiv and SVAMP**, following the experimental setup in \[3\], because the three datasets consist of similar arithmetic reasoning problems. We apologize for any confusion caused by our lack of clarity, and thank you for pointing out what we missed. We’ll add this to the text.
>
>
>
> ### 5. Grammatical correctness of remaining 71%
> * For the 29% of the questions that did not change, we used a grammar checking tool (Grammarly) to correct grammatical errors.
> * Afterward, all questions (# = 10,022) were checked for grammatical errors. This was done by three graduate students who were not involved in the translation and were proficient in English and Korean. The results showed that 5.24% (# = 525) of the questions had grammatical errors, which were corrected.
> * Due to the page limit, we had to compress the content, so the relevant information may not be clear. We apologize for the lack of explanation, and we will clarify this in the paper.
>
> ### References
> \[1\] OpenAI. GPT-4 Technical Report. ArXiv abs/2303.08774 (2023): n. pag.
> \[2\] Wei, Jason et al. Chain-of-Thought Prompting Elicits Reasoning in Large Language Models. NIPS (2022)
> \[3\] Gao, Luyu et al. PAL: Program-aided Language Models. ArXiv abs/2211.10435 (2022): n. pag.

---

### Official Review · Reviewer_jf9x · 2023-08-05

**Soundness:** 4

**Excitement:**

3: Ambivalent: It has merits (e.g., it reports state-of-the-art results, the idea is nice), but there are key weaknesses (e.g., it describes incremental work), and it can significantly benefit from another round of revision. However, I won't object to accepting it if my co-reviewers champion it.

**Paper Topic And Main Contributions:**

This paper introduces a new math word problem dataset. This purpose of this new dataset is to provide more diversity on languages, intermediate representations, and problem types, lexical usage patterns.

**Reasons To Accept:**

The dataset provides both Korean and English languages for math questions. To cover the diversity of math concepts, the dataset supports 5 problem types, including arithmetic, comparison, correspondence, geometry and possibility. It support two solution forms: express tree and python code. In expression tree, it provides 50 operators. Compared to SVAMP and ASDiv datasets, the experimental results show the difficulty of this dataset.

**Reasons To Reject:**

To show the difficulty of the dataset, it is compared with other datasets, such as GSM8K, SVAMP, ASDiv. The reason is now well analyzed, since those datasets are arithmetic types, and only contain basic arithmetic operators, but this dataset have more question types and operators. The Math Dataset contains similar question types as this dataset, but not compared.

**Reproducibility:**

3: Could reproduce the results with some difficulty. The settings of parameters are underspecified or subjectively determined; the training/evaluation data are not widely available.

**Reviewer Confidence:**

4: Quite sure. I tried to check the important points carefully. It's unlikely, though conceivable, that I missed something that should affect my ratings.

---

> ### Author Rebuttal · Authors · 2023-08-29
>
> We appreciate your insightful feedback and the detailed review of our paper. We are committed to addressing your questions and concerns and have responded below.
>
> ## Reasons to Reject
> ### 1. Comparison between DMath and MATH \[1\]
>
> * Initially, a concise summary of the responses is as follows:
>
>     1. **\[Discrepancy in domain knowledge levels\]**
>         We aimed to tackle mathematics problems at the elementary school level (grade 1-6), while MATH \[1\] is designed for the high school level (grade 8-12). Therefore, in our paper, we compared data at equivalent domain levels.
>     2. **\[General comparison between DMath and a subset of MATH\]**
>         Despite differences in domain knowledge levels and problem categories, creating a subset of MATH \[1\] similar to DMath types is feasible. As you suggested, the comparison between DMath and this subset of MATH \[1\] demonstrates that DMath is a dataset capable of evaluating fundamental reasoning skills.
>
> #### **\[Discrepancy in domain knowledge levels\]**
> * **Our objective was to develop an elementary school-level (grade 1-6) math dataset, whereas MATH \[1\] caters to high-school-level (grade 8-12) mathematics. To ensure fair comparison, we compared it with elementary school-level datasets such as GSM8K \[2\], SVAMP \[3\], and ASDiv \[4\].**
> * The MATH \[1\] dataset contains problems at the level of high school education. According to “Section 3.1. The MATH Dataset” (p. 4, lines 20-22) in \[1\], it encompasses intricate mathematical problems from AMC10, AMC12, and AIME that necessitate comprehension of advanced mathematical concepts such as trigonometric functions or imaginary numbers.
> * Additionally, we chose elementary-level math problems owing to their effectiveness in evaluating fundamental reasoning abilities. Although there are several Math Word Problem (MWP) datasets available at this level \[2, 3, 4\], they mainly concentrate on arithmetic reasoning. Nonetheless, examining the curriculum for these grades exposes other areas, such as relationships or probability, which also require an appraisal of reasoning skills. Consequently, **we designed DMath to provide a more comprehensive yet refined dataset that could assess these areas with efficacy.**
> * We briefly outlined these points in Appendix D, lines 957-972 and line 296. However, we will provide further detail on this particular point.
>
> #### **\[General comparison between DMath and a subset of MATH\]**
> * The MATH Dataset \[1\], comprising high-level questionnaires on domain-specific knowledge for grades 8-12, categorizes seven fields as described in "Section 3.1" (p. 4, lines 31-33 in \[1\]): Pre-algebra, Algebra, Number Theory, Counting and Probability, Geometry, Intermediate Algebra, and Precalculus.
> * **Regardless of the data's domain knowledge level, it is accepted that there may be types similar to the five types of DMath (Arithmetic, Comparison, Correspondence, Geometry, and Possibility) \[1\].** Therefore, we can categorize the seven types outlined in \[1\] as either similar to DMath or dissimilar.
>     - **Similar question types**
>         1. **PreAlgebra**: According to \[5\], it covers Exponents, Divisors, Greatest Common Divisor, Fractions, etc. **It is similar to the ARI \(Arithmetic\) category in DMath.**
>         2. **Counting & Probability**: According to \[6\], it contains Combinations, Basic Probability, Pascal's Triangle, Binomial Theorem, etc. **It is similar to Possibility (POS) in DMath.**
>         3. **Geometry**: According to \[7\], it contains Lines, Parallel and Perpendicular Lines, Right Triangles, Geometric Mean, Vectors, etc. **It is similar to Geometry (GEO) in DMath.**
>   - **Non-similar question types**
>     1. **Algebra**: According to \[8\], it contains contents such as Linear functions, Linear Inequalities, Quadratic Functions, etc.
>     2. **Intermediate Algebra**: According to \[9\], it includes Linear Equations, Polynomials, Exponential and Logarithmic Functions, etc.
>     3. **Precalculus**: According to \[10\], it includes Trigonometric, Matrices, Series, etc.
>     4. **Number Theory**: According to \[11\], it includes Primes, Fermat's Theorem, Fibonacci Numbers, etc.
>
> * **Table 1 displays the accuracy of PreAlgebra, Counting & Probability, and Geometry from MATH \[1\] in comparison to ARI, POS, and GEO from DMath.** The accuracy of GSM8K, ASDiv, and SVAMP, which comprise arithmetic reasoning, is also demonstrated. The result for MATH \[1\] in Table 1 (below) originates from Table 1 (p. 8) in \[12\]. The table below presents GPT-4 few-shot CoT performance.
>
> #### \<Table 1\>
>
> | DMath | Acc.  | MATH \[1\] | Acc. | GSM8K | Acc.  | ASDiv | Acc.  | SVAMP | Acc. |
> |-------|-------|------------------------|------|-------|-------|-------|-------|-------|------|
> | ARI   | 76.53 | PreAlagebra            | 71.6 |       | 91.81 |       | 90.03 |       | 91.6 |
> | POS   | 57.04 | Counting & Probability | 53.1 | -     | -     | -     | -     | -     | -    |
> | GEO   | 75.72 | Geometry               | 36.5 | -     | -     | -     | -     | -     | -    |
>
> * From the results displayed in Table 1, it is evident that **MATH \[1\], which demands an advanced level of domain knowledge (grade 8-12), predictably exhibits lower performance compared to the datasets that necessitate a lower level of domain knowledge (grade 1-6).**
> * Additionally, **DMath** exhibits a notably smaller performance disparity with MATH \[1\] compared to \[2, 3, 4\], **suggesting increased complexity of datasets within the same domain knowledge level.** DMath can **provide insights** that previous datasets do not because it **provides multiple reasoning categories at lower levels of domain knowledge (grades 1-6).**
>   * For example, POS does not perform well on low-level domain knowledge (grades 1-6), so we can see that even GPT-4, the current SOTA model, does not yet perform well on basic probability reasoning.
>   * On the other hand, GEO performs relatively well on low-level domain knowledge (grades 1-6), and about half as well on high-level domain knowledge (grades 8-12). This suggests that basic geometric reasoning is recognized by GPT-4 and that a lack of domain knowledge can lead to poor performance in MATH \[1\].
>   * Consequently, this means that DMath is currently the only dataset that can help researchers conduct mathematical reasoning tasks to assess the most basic reasoning abilities.
>
> * Finally, given that 1) they \[2, 3, 4\] are at the same domain knowledge level and 2) the types of MATH \[1\] and DMath are different, we thought it was fair to compare only DMath with \[2, 3, 4\]. However, a rough comparison suggests that comparing DMath with a subset of MATH \[1\] could be helpful for future math reasoning tasks. We are grateful for this suggestion, and we will consider it.
>
> ### References
> \[1\] Hendrycks et al. Measuring Mathematical Problem Solving With the MATH Dataset. NIPS (2021)
> \[2\] Cobbe, Karl et al. “Training Verifiers to Solve Math Word Problems.” ArXiv abs/2110.14168 (2021): n. pag.
> \[3\] Miao, Shen-Yun et al. “A Diverse Corpus for Evaluating and Developing English Math Word Problem Solvers.” ArXiv abs/2106.15772 (2020): n. pag.
> \[4\] Patel, Arkil et al. “Are NLP Models really able to Solve Simple Math Word Problems?” North American Chapter of the Association for Computational Linguistics (2021).
> \[5\] Rusczyk, Richard, et al. Prealgebra. 2011.
> \[6\] Patrick, David. Introduction to Counting and Probability. 2007.
> \[7\] Hill, McGraw. Geometry, Student Edition. McGraw-Hill Education, 2012.
> \[8\] Hill, McGraw. Algebra 1, Student Edition. McGraw-Hill Education, 2012.
> \[9\] Miller, Julie, et al. Intermediate Algebra. McGrawHill Education, 2022.
> \[10\] Miller, Julie, and Donna Gerken. Precalculus. 2022.
> \[11\] Burton, David. Elementary Number Theory. McGraw-Hill Education, 2010.
> \[12\] Zhou, Aojun et al. Solving Challenging Math Word Problems Using GPT-4 Code Interpreter with Code-based Self-Verification. ArXiv abs/2308.07921 (2023): n. pag.

---

### Official Review · Reviewer_ZcUJ · 2023-08-05

**Soundness:** 4

**Excitement:**

4: Strong: This paper deepens the understanding of some phenomenon or lowers the barriers to an existing research direction.

**Missing References:**

None.

**Paper Topic And Main Contributions:**

This paper proposes a new diverse dataset for math word problems. They collected 10K problems in English and Korean, including equations and Pythonic represnations to solve the problem. Also, they tested several PLMs on their dataset and showed that the math word problem solving task isn't over, LLM should handle some of the cases that their dataset presents.

**Questions For The Authors:**

Question A. Why does the finetuned model showed similar tendency to LLMs? Is there any further explanation about that? (lines 439-444)

**Reasons To Accept:**

- The paper is well written, and stating what they did.
- Providing a new dataset that ensures diversity on languages, linguistic patterns, and intermediate solution forms.
- Providing a thorough analysis of current LLMs on their datasets

**Reasons To Reject:**

Nothing.

**Reproducibility:**

4: Could mostly reproduce the results, but there may be some variation because of sample variance or minor variations in their interpretation of the protocol or method.

**Reviewer Confidence:**

4: Quite sure. I tried to check the important points carefully. It's unlikely, though conceivable, that I missed something that should affect my ratings.

**Typos Grammar Style And Presentation Improvements:**

- I think the size of GPT-4 model is stated in the GPT-4 technincal report.
- In lines 368, did you try to say "the worst problem categories differ across MWP modes"? Because, it's weird to compare problem categories with MWP models.
- In lines 414-418, the term "matched" means "are correctly solved"?

---

> ### Author Rebuttal · Authors · 2023-08-29
>
> We appreciate your feedback on our strengths. Below are our responses to your questions.
>
> ## Questions For the Authors
> ### 1. Different tendency between fine-tuned models and Large Language Models
> * Firstly, we aim to clarify the "tendency" in lines 439-444. **This tendency indicates that various forms of expressions have their preferred categories.** To elaborate, when we state that form of expression A outperforms form B, it does not imply that every type within A uniformly surpasses every type within B. Rather, it suggests that certain types within A outperform B, while some types in B excel over A. **This tendency is discernible in both fine-tuning and prompting methodologies.**
>
> * The fine-tuning and large language models (LLM) exhibit this inclination; however, **their reasons differ.** For the **fine-tuning model**, the length of the intermediate solution form significantly impacts preference determination. For instance, GEO—having the shortest code token length—performs more efficiently with Python code expression form, while COM and POS—with longer code token lengths—demonstrate superior performance with natural language expression forms.
>
> * Regarding **LLMs**, features intrinsic to the expression form greatly influence preference determination. Considerably lower mathematical computation errors distinguish Python code compared to natural language; conversely, natural language excels over Python code in succinctly expressing relationships between two objects on a large-small scale.
>
> * Lastly, we apologize for any ambiguity surrounding our use of 'tendency' in lines 439-444 and will incorporate a suitable explanation to clarify its meaning.
>
> ## Typos Grammar Style And Presentation Improvements
> ### 1. The size of the GPT-4 model
> * We have meticulously revisited the technical report \[1\] to ensure no details were overlooked. However, it does not provide any information regarding the parameter sizes of GPT-4. As stated in the "2. Scope and limitations of this Technical Report" section (p.2, lines 13-16), it omits specifics of model size, the hardware used, computational training power, dataset construction, training methodology, etc. Should information about parameter sizes become publicly accessible, we will certainly incorporate it.
>
> ### 2. In line 368, ‘across’ seems more suitable than ‘from.’
> * Your suggestion is indeed valuable. As you rightly pointed out, using "across" instead of "from" suits our intended message here. Our objective was not to draw comparisons between models and categories but to illustrate that different models exhibit varying degrees of performance across various categories. As per your recommendation, we will employ "across" for a clearer expression of our standpoint. We appreciate your insightful feedback on this matter.
>
>
>
> ### 3. In lines 414-418, the term "matched" means "are correctly solved"?
> * You raise a valid observation. The subsequent line presents the outcome of classifying the questions resolved utilizing both Natural Language (NL) prompts and Python code prompts into four distinct scenarios:
>     1. The proportion of questions correctly addressed solely through NL prompts
>     2. The proportion of questions correctly addressed using both NL and Python code prompts
>     3. The proportion of questions correctly addressed exclusively via Python code prompts
>     4. The proportion of questions that remained incorrect despite employing both methods
>
> * We have employed the term **"matched"** to denote inclusion within any of the four scenarios as mentioned earlier. However, as you rightly pointed out, it would be more precise to phrase it as "are accurately resolved" in this context. We are grateful for your point and will implement this amendment.
>
> ### References
> \[1\] OpenAI. GPT-4 Technical Report. ArXiv abs/2303.08774 (2023): n. pag.

---

### Meta-Review · Area_Chair_X5t7 · 2023-09-12

**Recommendation:** 5

**Metareview:**

The paper’s major contribution is a novel math word problem dataset in English and Korean characterized by diversity at various levels: in problem types, in lexical patterns, in language narratives, in solution forms, which can help better evaluate the actual linguistic and reasoning performance of LLMs. The authors conducted experiments with various pre-trained language models and found that the math word problem-solving task remains challenging. Reviewers converged on soundness and found the work novel and reproducible. Overall, the paper does not show major weaknesses and may improve results and findings in this area.

**Pros.**

- The paper is well written, and argumentation is sound and clear;

- A novel gold dataset of math word problem of higher complexity and diversity on various aspects;

- The dataset addresses gaps in previous works and has the potential to enhance future research in the field;

- The methodology seems transferable to other languages.

**Cons:**

No a serious weaknesses to highlight. However, the rationale of the choices made for the comparison with other data sets could be more clearly explained/ motivated.

---

### Decision · Program_Chairs · 2023-10-07

**Decision:**

Accept-Main

**Comment:**

The paper’s major contribution is a novel math word problem dataset in English and Korean characterized by diversity at various levels: in problem types, in lexical patterns, in language narratives, in solution forms, which can help better evaluate the actual linguistic and reasoning performance of LLMs. The authors conducted experiments with various pre-trained language models and found that the math word problem-solving task remains challenging. Reviewers converged on soundness and found the work novel and reproducible. Overall, the paper does not show major weaknesses and may improve results and findings in this area.

**Pros.**

- The paper is well written, and argumentation is sound and clear;

- A novel gold dataset of math word problem of higher complexity and diversity on various aspects;

- The dataset addresses gaps in previous works and has the potential to enhance future research in the field;

- The methodology seems transferable to other languages.

**Cons:**

No a serious weaknesses to highlight. However, the rationale of the choices made for the comparison with other data sets could be more clearly explained/ motivated.